# A Theoretical Study on Bridging Internal Probability and Self-Consistency for LLM Reasoning

**Zhi Zhou** [1]   **Yuhao Tan** [1]   **Zenan Li** [2]   **Yuan Yao** [1]
**Lan-Zhe Guo** [1,3]   **Yu-Feng Li** [1,4*]   **Xiaoxing Ma** [1*]

[1]State Key Laboratory of Novel Software Technology, Nanjing University, China
[2]Department of Computer Science, ETH Zurich, Switzerland
[3]School of Intelligence Science and Technology, Nanjing University, China
[4]School of Artifical Intelligence, Nanjing University, China
zhouz@lamda.nju.edu.cn, liyf@nju.edu.cn, xxm@nju.edu.cn

## Abstract

Test-time scaling seeks to improve the reasoning performance of large language models (LLMs) by adding computational resources. A prevalent approach within the field is *sampling-based test-time scaling methods*, which enhance reasoning by generating multiple reasoning paths for a given input during inference. However, despite its practical success, the theoretical foundations remain underexplored. In this paper, we provide the first theoretical framework for analyzing sampling-based test-time scaling methods, grounded in the perspective of confidence estimation. Based on the framework, we analyze two dominant paradigms: self-consistency and perplexity, and reveal key limitations: self-consistency suffers from high estimation error while perplexity exhibits substantial modeling error and possible degradation of the estimation error convergence. To address these limitations, we introduce RPC, a hybrid method that leverages our theoretical insights through two key components: *Perplexity Consistency* and *Reasoning Pruning*. *Perplexity Consistency* combines the strengths of self-consistency and perplexity, boosting the convergence rate of estimation error from linear to exponential while preserving model error. *Reasoning Pruning* prevents degradation by eliminating low-probability reasoning paths. Both theoretical analysis and empirical results across seven benchmark datasets demonstrate that RPC has a strong potential for reducing reasoning error. Notably, RPC achieves reasoning performance comparable to self-consistency while not only enhancing confidence reliability but also reducing sampling costs by 50%. The code and resources are available at `https://wnjxyk.github.io/RPC`.

## 1   Introduction

Recent advances in large language models (LLMs) have demonstrated their remarkable reasoning capabilities across diverse applications, including problem-solving [33, 35], planning [53, 13], and decision-making [45, 48]. Test-time scaling methods [58, 57, 20] can further enhance reasoning performance with additional computation. Among these strategies, the sampling-based test-time scaling method has emerged as a simple yet effective technique. Through generating multiple reasoning paths and selecting the most plausible one through a confidence estimation mechanism, such as self-consistency [57] and perplexity [10], this approach has achieved significant improvements.

The success of sampling-based test-time scaling hinges on accurately estimating the confidence of sampled reasoning paths, facilitating their comparison and enabling the selection of the most

---

[*]Corresponding author.

plausible answer in the Best-of-N manner [20, 29]. Current confidence estimation approaches can be categorized into two main types: (i) *consistency-based methods*, which use pre-defined consistency functions [11] to determine answer confidence, with self-consistency as the primary representative; and (ii) *probability-based methods*, which utilize scores from either the internal LLM probability [44] or external model [29] to evaluate the confidence of reasoning paths, with perplexity as the leading example. While both have proven effective in practice, a rigorous theoretical understanding remains lacking regarding their underlying mechanisms, inherent limitations, and potential improvements.

This paper addresses this gap by introducing the first theoretical framework for sampling-based test-time scaling methods in LLM reasoning. Within this framework, reasoning error is decomposed into two components: *Estimation Error* and *Model Error*. We apply our framework to analyze two representative methods from each category, self-consistency and perplexity, and identify their limitations. Self-consistency, which relies on Monte Carlo estimation, achieves only a linear convergence rate in estimation error, resulting in unsatisfactory performance when the sampling budget is limited. Perplexity utilizes the internal probabilities of LLMs to achieve exponential convergence rates, but they suffer from high model error, which compromises their effectiveness. Moreover, its convergence rate depends on the magnitude of probabilities and deteriorates significantly when probabilities are low. These insights provide guidance for developing improved methods: *An optimal approach should simultaneously achieve rapid estimation error convergence while maintaining low model error*.

To address the above limitations, we introduce a novel **Reasoning-pruning Perplexity Consistency** (RPC) method, which consists of two components: *Perplexity Consistency* and *Reasoning Pruning*. *Perplexity Consistency* integrates the internal probability of the LLM into the self-consistency framework, enabling rapid estimation error reduction that transitions from linear to exponential convergence while maintaining low model error. *Reasoning Pruning* mitigates the degradation issue of estimation error reduction rate when the internal LLM probability magnitude is low by automatically modeling the probability distribution and removing low-probability reasoning paths. Our theoretical analysis shows that RPC achieves both fast estimation error convergence and low model error, offering significant potential to reduce reasoning error. Our empirical results demonstrate the effectiveness and efficiency of RPC on seven benchmark datasets. Specifically, on four mathematical reasoning datasets, RPC reduces the required sampling budget by at least 50% while maintaining the same level of reasoning performance as self-consistency. When using the same sampling budget, RPC achieves an average improvement of 1.29% over existing methods. Additionally, RPC provides confidence estimates that are more closely aligned with the ground truth compared to existing methods.

To summarize, the main contributions of the paper are:

(1) We propose a theoretical framework that formulates sampling-based test-time scaling in LLM reasoning and decomposes reasoning error into estimation error and model error. This framework enables an analysis of existing techniques, offering guidance for developing improved approaches.

(2) Based on this framework, we introduce a novel test-time scaling method, RPC, which leverages the internal probability of LLMs within the self-consistency paradigm and removes low-probability reasoning paths, thereby accelerating error reduction and improving reasoning performance.

(3) Our theoretical analysis demonstrates that RPC achieves rapid estimation error convergence, providing strong potential for reducing reasoning error. Empirical results on seven benchmark datasets show that RPC substantially reduces the required sampling budget, improves reasoning accuracy, and enhances the reliability of confidence estimation.

## 2 Problem and Analysis

In this section, we first formulate the LLM reasoning problem using sampling-based test-time scaling methods. Then, we theoretically decompose the LLM reasoning error into *Estimation Error* and *Model Error*, and analyze two representative methods of consistency-based and probability-based methods. Our analysis reveals insights for building an advanced LLM reasoning method.

### 2.1 Problem Formulation

Given a reasoning problem $(x, y)$, where $x$ represents the input query, and $y$ represents the ground-truth answer. The LLM generates a reasoning path $t = (t_1, \ldots, t_m)$ by sequentially sampling tokens

according to the conditional probability distribution $p(t_i \mid x, t_{<i})$, where $m$ denotes the length of the reasoning path. The probability of generating the reasoning path $\hat{t}$ is defined as $p(\hat{t} \mid x)$, a.k.a the confidence of the reasoning path $\hat{t}$. An extraction function $g(\cdot)$ maps the reasoning path to the final answer $\hat{y} = g(\hat{t})$. We can further extend the probability to the answer $\hat{y}$, i.e., the answer confidence, denoted as $p(\hat{y} \mid x)$. Take the mathematical reasoning problem as an example, the reasoning path $\hat{t}$ could be "1 + 1 = 2. The answer is 2." and the function $g(\cdot)$ extracts the answer from $\hat{t}$, resulting in $\hat{y} = g(\hat{t}) = 2$. The reasoning correctness is evaluated by the indicator function $\mathbb{I}[\hat{y} = y]$.

The confidence represents the probability that the reasoning path $\hat{t}$ or answer $\hat{y}$ is correct, allowing LLMs to select the most reliable solution from multiple candidates in the Best-of-N manner [20, 29]. However, in practice, accessing the confidence for all possible reasoning paths or answers of LLMs is computationally infeasible. Therefore, we typically estimate the confidence by sampling finite $n$ reasoning paths $\tilde{t}_1, \ldots, \tilde{t}_n$ from the LLM sampling distribution $p(t \mid x)$, which yields the estimated confidence $\hat{p}(\hat{t} \mid x)$ or $\hat{p}(\hat{y} \mid x)$ for any reasoning path $\hat{t}$ or answer $\hat{y}$.

To measure the reasoning performance of LLMs, we use the squared error [22] to penalize reasoning error of confidence for any single reasoning path $\hat{t}$ or answer $\hat{y}$:

$$\mathcal{E}_{\hat{p}}(\hat{t}) = \mathbb{E}\left[\left(\hat{p}(\hat{t} \mid x) - \mathbb{I}[g(\hat{t}) = y]\right)^2\right], \quad \mathcal{E}_{\hat{p}}(\hat{y}) = \mathbb{E}\left[\left(\hat{p}(\hat{y} \mid x) - \mathbb{I}[\hat{y} = y]\right)^2\right]. \quad (1)$$

where the expectation is taken over all possible combinations of $n$ sampled reasoning paths $\tilde{t}_1, \ldots, \tilde{t}_n$, which are used to estimate the confidence $\hat{p}$. For notational clarity and simplicity, we omit the explicit form of expectation here and throughout the remainder of the paper.

In the following analysis, we categorize sampling-based test-time scaling methods into two types based on confidence estimation aspects: consistency-based methods and probability-based methods. Consistency-based methods [63, 72] estimate confidence by evaluating the agreement among different reasoning paths, with self-consistency [57] serving as a representative approach. Probability-based methods [20, 10] estimate confidence using either the internal probability provided by the LLM or the scores from external models, with perplexity [57] being a representative example. Our further analysis is conducted on each representative method.

## 2.2 Theoretical Analysis

In this section, we theoretically decompose the reasoning error into estimation error and model error. Next, we examine the specific forms of reasoning error for two representative methods, self-consistency (SC) and perplexity (PPL), to provide insights for improved algorithm design.

### 2.2.1 Reasoning Error Decomposition

Take the reasoning error $\mathcal{E}_{\hat{p}}(\hat{y})$ of $\hat{p}(\hat{y} \mid x)$ in Equation 1 as an example, we can decompose the reasoning error into the estimation error and model error as follows.

**Proposition 1** (Error Decomposition). *For any input $x$ with ground-truth answer $y$ and any possible answer $\hat{y}$, let $\hat{p}(\hat{y} \mid x)$ denote the unbiased estimated confidence of $\hat{y}$ and $p(\hat{y} \mid x)$ denote the ground truth confidence. Then, the reasoning error $\mathcal{E}_{\hat{p}}(\hat{y})$ can be divided into two components:*

$$\mathcal{E}_{\hat{p}}(\hat{y}) = \underbrace{\mathbb{E}\left[\left(\hat{p}(\hat{y} \mid x) - p(\hat{y} \mid x)\right)^2\right]}_{\text{Estimation Error}} + \underbrace{\left(p(\hat{y} \mid x) - \mathbb{I}[\hat{y} = y]\right)^2}_{\text{Model Error}}, \quad (2)$$

*where the expectation is taken over sampled reasoning paths $\tilde{t}_1, \ldots, \tilde{t}_n$ for estimating confidence.*

*Remark* 1. The detailed proof is provided in Appendix A.1. Proposition 1 separates the effect of confidence estimation on reasoning error from the effect of the LLM's reasoning capability. The *Estimation Error* depends solely on the sampling size and the confidence estimation strategy, while the *Model Error* is invariant and determined by the LLM's reasoning capability. This proposition demonstrates that, apart from the fixed *Model Error*, which is determined by the LLM's inherent reasoning capability, the reasoning error is bounded by the *Estimation Error*. Moreover, this proposition provides insights into two directions for improving LLM reasoning performance: (1) reducing the *Estimation Error* through larger sampling sizes or more accurate confidence estimation methods, and (2) reducing the *Model Error* by enhancing the LLM's reasoning capabilities or developing more sophisticated and effective confidence metrics.

Next, we analyze two representative methods in sampling-based test-time scaling: self-consistency (SC) from consistency-based methods and perplexity (PPL) from probability-based methods, using our theoretical framework. Below, we adopt a common assumption that LLM sampling follows a Bernoulli distribution [55], allowing us to compute the estimation error for specific methods.

### 2.2.2 Analysis of Self-Consistency

Self-consistency [60, 1, 4] is the representative method for consistency-based methods, which samples $n$ reasoning paths $\tilde{t}_1, \dots, \tilde{t}_n$, and estimates the confidence of any $\hat{y}$ using Monte-Carlo estimation by

$$\hat{p}^{(\text{SC})}(\hat{y} \mid x) = \frac{1}{n} \sum_{i=1}^{n} \mathbb{I}[\tilde{y}_i = \hat{y}], \quad \tilde{y}_i = g(\tilde{t}_i). \tag{3}$$

Then, the reasoning error of SC for a given problem $(x, y)$ and any possible $\hat{y}$ can be computed by

$$\mathcal{E}_{\hat{p}^{(\text{SC})}}(\hat{y}) = \mathbb{E}\left[ \left( \frac{1}{n} \sum_{i=1}^{n} \mathbb{I}[\tilde{y}_i = \hat{y}] - \mathbb{I}[\hat{y} = y] \right)^2 \right]. \tag{4}$$

Finally, we conduct decomposition on SC to illustrate the key factors affecting the reasoning error.

**Proposition 2** (SC Reasoning Error Decomposition). *For any input $x$ with ground-truth answer $y$, let $\hat{p}^{(\text{SC})}(\hat{y} \mid x)$ denote the estimated probability for any possible answer $\hat{y}$ by SC. Then, the reasoning error $\mathcal{E}_{\hat{p}^{(\text{SC})}}(\hat{y})$ can be divided into two components:*

$$\mathcal{E}_{\hat{p}^{(\text{SC})}}(\hat{y}) = \underbrace{\frac{1}{n} p(\hat{y} \mid x)(1 - p(\hat{y} \mid x))}_{\text{Estimation Error}} + \underbrace{\left( p(\hat{y} \mid x) - \mathbb{I}[\hat{y} = y] \right)^2}_{\text{Model Error}}. \tag{5}$$

*Remark* 2. The detailed proof is provided in Appendix A.1. This proposition reveals that the estimation error of SC consists solely of variance since the sampling is unbiased. It decreases only linearly with increasing sample size, which results in substantial reasoning error when sampling is limited. This analysis suggests that a promising direction for improving SC is to develop methods that achieve faster estimation error convergence rates.

### 2.2.3 Analysis of Perplexity

Perplexity is a representative method for probability-based methods that directly utilizes the internal LLM probability $p(\hat{t} \mid x)$ for any reasoning path $\hat{t}$. However, since the number of possible reasoning paths is nearly infinite, the ground-truth probability can only be accessed for those paths that are actually sampled. Therefore, the estimated probability of any possible reasoning path $\hat{t}$ is estimated using the unique set of $n$ sampled reasoning paths $\mathcal{R} = \text{Set}\left( \tilde{t}_1, \dots, \tilde{t}_n \right)$.

$$\hat{p}^{(\text{PPL})}(\hat{t} \mid x) = \begin{cases} p(\tilde{t}_i \mid x), & \text{if } \hat{t} = \tilde{t}_i, \\ 0, & \text{otherwise.} \end{cases} = \sum_{\tilde{t} \in \mathcal{R}} \mathbb{I}\left[ \hat{t} = \tilde{t} \right] p(\tilde{t} \mid x). \tag{6}$$

Similarly, the reasoning error of PPL is denoted as follows, with the following proposition decomposing the reasoning error of PPL.

$$\mathcal{E}_{\hat{p}^{(\text{PPL})}}(\hat{t}) = \mathbb{E}\left[ \left( \sum_{\tilde{t} \in \mathcal{R}} \mathbb{I}\left[ \hat{t} = \tilde{t} \right] p(\tilde{t} \mid x) - \mathbb{I}[g(\hat{t}) = y] \right)^2 \right]. \tag{7}$$

**Proposition 3** (PPL Reasoning Error Decomposition). *For any given input $x$ with ground-truth answer $y$, let $\hat{p}^{(\text{PPL})}(\hat{t} \mid x)$ denote the estimated probability of $\hat{t}$ by PPL method for any possible reasoning path $\hat{t}$. Then, the reasoning error $\mathcal{E}_{\hat{p}^{(\text{PPL})}}(\hat{t})$ can be divided into two components:*

$$\mathcal{E}_{\hat{p}^{(\text{PPL})}}(\hat{t}) = \underbrace{(1 - p(\hat{t} \mid x))^n p(\hat{t} \mid x)(2\mathbb{I}[\hat{y}_i = y] - p(\hat{t} \mid x))}_{\text{Estimation Error}} + \underbrace{\left( p(\hat{t} \mid x) - \mathbb{I}[g(\hat{t}) = y] \right)^2}_{\text{Model Error}}. \tag{8}$$

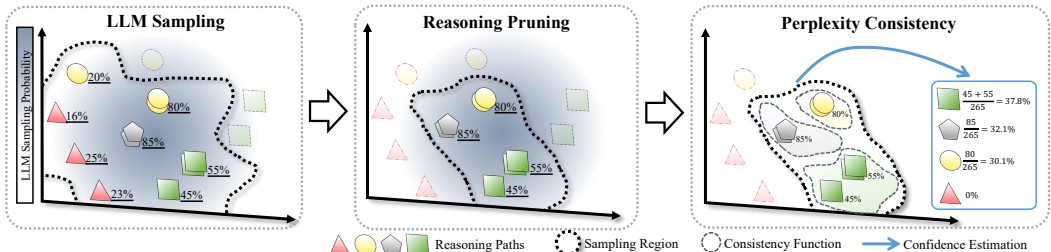

Figure 1: Illustration of the RPC approach. The *Reasoning Pruning* filters out low-probability answers, while the *Perplexity Consistency* incorporates LLM probabilities into the self-consistency framework, resulting in faster convergence of estimation error.

*Remark* 3. The detailed proof is provided in Appendix A.1. Compared with SC, the estimation error of PPL decreases exponentially, with the rate depending on the magnitude of ground-truth confidence $p(\hat{t} \mid x)$. For reasoning paths that are likely to yield the correct answer (i.e., those with non-negligible confidence), PPL achieves a substantially faster convergence rate. However, for reasoning paths with extremely low confidence, the convergence advantage of PPL degrades. Furthermore, the *Model Error* of PPL is typically larger than that of SC in practice, and we formally demonstrate this in the ideal case in Appendix 4. This analysis suggests that the degradation issue of *Estimation Error* and the large *Model Error* issue are fundamental challenges for improving PPL method.

## 3 Methodology

Based on our theoretical analysis, we identify three fundamental challenges in existing methods: (1) SC suffers from slow *Estimation Error* convergence, leading to efficiency concerns; (2) PPL exhibits large *Model Error*, compromising its effectiveness; (3) PPL's *Estimation Error* convergence advantage deteriorates significantly for certain cases, resulting in a degradation issue.

To address these challenges, we propose *Reasoning-Pruning Perplexity Consistency* (RPC), a novel method with two key components. First, we integrate internal LLM probabilities into the self-consistency framework to create *Perplexity Consistency* (PC), a confidence estimation function that reduces estimation error efficiently while maintaining low model error, thus addressing the first two challenges with theoretical guarantees. Second, we introduce a *Reasoning Pruning* (RP) module that resolves the third challenge by systematically filtering out reasoning paths with low probabilities.

### 3.1 Perplexity Consistency

To address the efficiency and effectiveness challenges simultaneously, we propose PC, which directly leverages the LLM's prediction probability like PPL, obtaining the benefit of an exponential convergence rate, and also applies the consistency function of SC to minimize the model error. Formally, for the unique set of $n$ sampled reasoning paths $\mathcal{R} = \mathrm{Set}\left(\tilde{t}_1, \ldots, \tilde{t}_n\right)$, the estimated probability of any possible answer $\hat{y}$ is

$$\hat{p}^{(\mathrm{PC})}(\hat{y} \mid x) = \sum_{\tilde{t} \in \mathcal{R}} \mathbb{I}[g(\tilde{t}) = \hat{y}] p(\tilde{t} \mid x), \tag{9}$$

which calculates the cumulative probability of all unique reasoning paths $\left\{\tilde{t}_i \mid \tilde{t}_i \in \mathcal{R} \text{ and } g(\tilde{t}_i) = \hat{y}\right\}$ whose answer is $\hat{y}$. Therefore, the squared error of PC for any possible answer $\hat{y}$ is

$$\mathcal{E}_{\hat{p}^{(\mathrm{PC})}}(\hat{y}) = \mathbb{E}\left[(\hat{p}^{(\mathrm{PC})}(\hat{y} \mid x) - \mathbb{I}[\hat{y} = y])^2\right]. \tag{10}$$

Now, we present the following theorem, which explores the reasoning error decomposition of PC.

**Theorem 4** (PC Reasoning Error Decomposition). *Assume that* $k = |\{\tilde{t} \mid g(\tilde{t}) = \hat{y}\}|$ *and define* $\alpha := 1 - \frac{1}{k} p(\hat{y} \mid x)$. *Then, the reasoning error* $\mathcal{E}(\hat{p}^{(\mathrm{PC})})$ *of* PC *can be divided into two components:*

$$\mathcal{E}_{\hat{p}^{(\mathrm{PC})}}(\hat{y}) = \underbrace{\alpha^n p(\hat{y} \mid x)\left(2\mathbb{I}[\hat{y} = y] - (1 + \alpha^n) p(\hat{y} \mid x)\right)}_{\textit{Estimation Error}} + \underbrace{(p(\hat{y} \mid x) - \mathbb{I}[\hat{y} = y])^2}_{\textit{Model Error}}.$$

*Remark* 5. The proof is presented in Appendix A.3. The theorem states that PC successfully fuses the strengths of PPL and SC: it achieves the same level of model error as SC while ensuring the similar convergence rate as PPL in the estimation error. Particularly, the convergence rate can be computed as $\alpha^n p(\hat{y} \mid x) = (1 - \frac{1}{k}p(\hat{y} \mid x))^n p(\hat{y} \mid x)$, which is exponential.

*Remark* 6. The convergence rate is primarily influenced by the magnitude of $p(\hat{y} \mid x)$. In most scenarios, it remains exponential, facilitating rapid estimation error reduction. However, when $p(\hat{y} \mid x) \to 0$ and $np(\hat{y} \mid x) \ll 1$, we only have $\alpha^n \to \frac{1}{1+np(\hat{y} \mid x)}$ [32], resulting in the convergence rate unexpectedly degenerating to a linear result.

## 3.2 Reasoning Pruning

To address the third degradation challenge, which is also verified in Remark 6, we propose directly pruning these sampled answers, called *Reasoning Pruning* (RP). The main idea of RP is that an answer with negligible $p(\hat{y} \mid x)$ is not likely to be the correct answer, so we may prune it directly without a significant increase in reasoning error. Specifically,RP sets $\hat{p}(\hat{y} \mid x) = 0$ when the cumulative probability of all its corresponding reasoning paths is lower than a threshold $\tau$, making the estimation error vanish. However, there are two questions raised for RP: (1) Although pruning can boost the estimation error reduction, does it potentially increase the model error and thereby increase the overall reasoning error? (2) How should we determine the threshold $\tau$ for pruning?

For the first question, we theoretically prove that RP can achieve the optimal error reduction for each given problem $(x, y)$ with the optimal threshold $\tau$ at a high probability.

**Theorem 7** (Effectiveness of Reasoning Path Pruning). *Assume that the optimal threshold* $\tau = p(y \mid x)$, *and let* $\hat{k} = |\{\tilde{t}_i \mid g(\tilde{t}_i) = \hat{y}, i = 1, \dots, n\}|$, *which refers to the size of samples whose answer is* $\hat{y}$. *Hence,* RP *achieves the optimal error reduction with at least the probability*

$$1 - \exp\left(-2\hat{k}k^2(1 - \frac{\tau}{1-\alpha})^2\right).$$

*Remark* 8. The proof is included in Appendix A.4. The theorem provides a guarantee that RP can achieve the optimal error reduction for each given problem $(x, y)$ at a high probability. Note that the optimal error reduction not only boosts the estimation error reduction efficiency but also effectively reduces the model error, thus improving the final reasoning capability of LLMs.

For the second question, we develop an automated strategy to determine the $\tau$ based on the distribution of all sampled reasoning paths. Inspired by open-set recognition [5], we model the probability distribution of $\Omega_1$ and $\Omega_2$ as a mixture of two Weibull distributions, representing high and low probability regions. Elaborately, we define the PDF of the mixture distribution as:

$$f(x) = w_1 f_{\mathrm{W}}(x; k_1, \lambda_1) + w_2 f_{\mathrm{W}}(x; k_2, \lambda_2),$$

where the Weibull PDF [59] is defined as $f_{\mathrm{W}}(x; k, \lambda) = \frac{k}{\lambda}\left(\frac{x}{\lambda}\right)^{k-1}\exp\left(-\left(\frac{x}{\lambda}\right)^k\right)$. We use the maximum likelihood estimation methods to estimate the parameters, i.e., $(k_1, \lambda_1)$, $(k_2, \lambda_2)$, $w_1$, and $w_2$ on the probability distribution of all sampled reasoning paths for each reasoning problem. We assume that Weibull$(k_1, \lambda_1)$ is the high probability distribution and Weibull$(k_2, \lambda_2)$ is the other. Then, the probability of the reasoning path $\hat{t}$ being in the high probability distribution is derived by

$$P_{\mathrm{High}}(x) = \frac{w_1 f_{\mathrm{W}}(x; k_1, \lambda_1)}{w_1 f_{\mathrm{W}}(x; k_1, \lambda_1) + w_2 f_{\mathrm{W}}(x; k_2, \lambda_2)}.$$

where $x$ is the value of LLM internal probability. Then, we remove sampled reasoning paths $\tilde{t}$ satisfying $P_{\mathrm{High}}(\hat{p}(\tilde{t} \mid x)) < 0.5$, which are more likely to be in the low probability distribution. Moreover, to ensure the algorithm's stability when $n$ is limited, we employ the Truncated Mean method [43], retaining outputs with a probability greater than the overall mean. This prevents the removal of too many reasoning paths due to the potential inaccurate estimation of the mixture distribution.

## 3.3 RPC Method

Overall, we apply the *Reasoning Pruning* to all sampled reasoning paths $\tilde{t}_1, \dots, \tilde{t}_n$ for removing low probability reasoning paths and then compute the confidence based on *Perplexity Consistency*, forming our proposed **R**easoning-pruning **P**erplexity **C**onsistency (RPC) confidence estimation method. The pseudo-code is shown in Algorithm 1 in Appendix B. Figure 1 illustrates its complete framework.

Table 1: Efficiency comparison of *Perplexity Consistency* module (Pc) and Rpc. The table shows the minimum number of samples needed to exceed the best performance of Sc, with reduction rates in bold when sampling is reduced.

| Method | MATH | | MathOdyssey | | OlympiadBench | | AIME | |
|---|---|---|---|---|---|---|---|---|
| | Accuracy | #Samplings | Accuracy | #Samplings | Accuracy | #Samplings | Accuracy | #Samplings |
| Best of Sc | 50.57 | 64 | 28.32 | 112 | 11.07 | 128 | 9.40 | 128 |
| Pc | 50.63 | 32 | 28.51 | 112 | 11.07 | 128 | 9.00 | 64 |
| Δ | +0.06 | **-50.0%** | +0.19 | -0.0% | 0.00 | -0.0% | 0.00 | **-50.0%** |
| Rpc | 51.16 | 32 | 29.31 | 32 | 11.07 | 64 | 9.50 | 48 |
| Δ | +0.59 | **-50.0%** | +0.99 | **-71.4%** | 0.00 | **-50.0%** | +0.10 | **-62.5%** |

## 4 Experiments

In this section, we conduct experiments to answer the following research questions:

**RQ1: Efficiency.** How does Rpc reduce the number of samples required to achieve comparable performance through faster convergence?

**RQ2: Efficacy.** How does Rpc improve reasoning performance compared to existing methods?

**RQ3: Reliability.** How does Rpc enhance the reliability of confidence estimation compared to existing methods?

Moreover, further discussions are devoted to further demonstrating the effectiveness of Rpc.

### 4.1 Experimental Setting

In this section, we briefly introduce the comparison methods, datasets, and implementation details. Due to space limitations, detailed experimental settings are included in Appendix C.

**Comparison Methods.** We compare three types of LLM confidences: perplexity confidence [57] (Ppl), self-consistency confidence [10] (Sc), and verbalized confidence [52] (Verb). The verbalized confidence is computed based on the probability that the LLM outputs "True" versus "False" when asked an "Is-True" question. For code generation tasks, we extracted verbalized confidence scores from the model's numerical likelihood expressions by prompting the LLM.

**Datasets.** We introduce four popular benchmarks for math reasoning: MATH [25], MathOdyssey [16], OlympiadBench [23], and AIME [67] (contains problems from 1983 to 2024). As to code generation tasks, we evaluate each method on three benchmarks, i.e., HumanEval [8], MBPP [3], and introductory-level problems of APPS [24].

**Implementation Details.** For math reasoning tasks, we evaluate the InternLM2-Math-Plus models with 1.8B and 7B parameters [65], as well as the DeepSeekMath-RL 7B model [49]. The consistency function $\mathbb{I}_C$ is the answer comparison. For code generation tasks, we evaluate the Deepseek-Coder 33B model. The consistency function $\mathbb{I}_C$ is constructed based on semantic equivalence [42] by clustering code based on given test cases. We set the sample size to $n = 128$ for the MathOdyssey, OlympiadBench, and AIME datasets and $n = 64$ for the MATH dataset by default. Each experiment is repeated 10 times with different random seeds, and the average performance is reported. All experiments were conducted on Linux servers with A800 and H800 GPUs.

### 4.2 Empirical Results

**RQ1: Efficiency.** How does Rpc reduce the number of samples required to achieve comparable performance through faster convergence?

We evaluate our proposed Rpc against the standard self-consistency method using four mathematical benchmark datasets with the InternLM-2-MATH-Plus 7B model. For the MATH dataset, we set the reasoning path size to 64, while we set the number of reasoning paths to 128 for the other datasets with Sc. We then record the best performance and minimum sampling requirements for Sc. For both

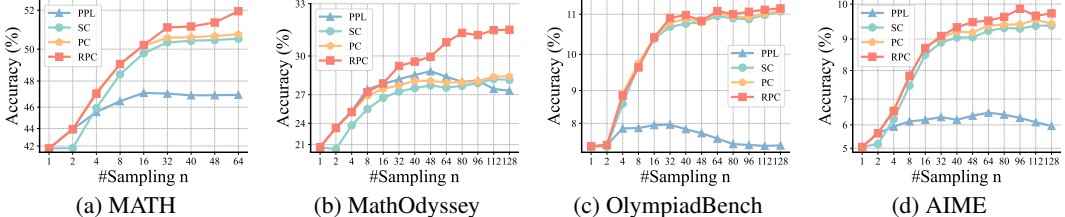

(a) MATH  (b) MathOdyssey  (c) OlympiadBench  (d) AIME

Figure 2: The accuracy of the InternLM-2-MATH-Plus 7B model on four math reasoning datasets with different sample sizes $n$. Our proposed RPC achieves the best performance across all datasets, validating our theoretical analysis. We also show the performance of a single perplexity consistency module (denoted as PC), which further supports our theoretical findings.

Table 2: Performance Comparison using InternLM-2-MATH-Plus 7B model measured by accuracy and expected calibration error metrics. The best performance is highlighted in **bold**. The results show that our RPC outperforms existing methods in majority of cases.

| Method | MATH | | MathOdyssey | | OlympiadBench | | AIME | | Average | |
|---|---|---|---|---|---|---|---|---|---|---|
| | Accuracy(↑) | ECE(↓) | Accuracy(↑) | ECE(↓) | Accuracy(↑) | ECE(↓) | Accuracy(↑) | ECE(↓) | Acc.(↑) | ECE(↓) |
| PPL | $46.99 \pm 0.20$ | $48.99 \pm 0.19$ | $27.35 \pm 1.22$ | $67.70 \pm 1.22$ | $7.27 \pm 0.36$ | $86.90 \pm 0.35$ | $5.96 \pm 0.48$ | $88.98 \pm 0.49$ | 21.90 | 73.14 |
| VERB | $26.14 \pm 0.25$ | $47.46 \pm 0.07$ | $10.06 \pm 0.61$ | $69.92 \pm 0.88$ | $3.68 \pm 0.16$ | $84.68 \pm 0.25$ | $3.17 \pm 0.17$ | $86.29 \pm 0.20$ | 10.76 | 72.09 |
| SC | $50.57 \pm 0.17$ | $6.71 \pm 0.18$ | $28.25 \pm 0.60$ | $12.23 \pm 0.54$ | $11.07 \pm 0.15$ | $20.20 \pm 0.16$ | $9.40 \pm 0.21$ | $14.35 \pm 0.23$ | 24.82 | 13.37 |
| RPC | $\mathbf{51.95 \pm 0.15}$ | $\mathbf{6.41 \pm 0.18}$ | $\mathbf{31.62 \pm 0.75}$ | $\mathbf{9.87 \pm 0.73}$ | $\mathbf{11.14 \pm 0.15}$ | $\mathbf{18.86 \pm 0.18}$ | $\mathbf{9.74 \pm 0.23}$ | $\mathbf{14.32 \pm 0.21}$ | **26.11** | **12.37** |

RPC and our *Perplexity Consistency* module (denoted as PC), we report the minimum number of samples needed to match or exceed the performance of the SC in Table 1.

The results of PC indicate improved convergence rates compared to SC in several cases, while maintaining similar rates in others. These findings support the rapid convergence and degeneration issues of PC in Theorem 4. RPC shows significant efficiency improvements by requiring fewer samples to achieve comparable performance relative to SC. Moreover, the degeneration issues observed in PC are effectively addressed in RPC, validating both the effectiveness of the *Reasoning Pruning* module and our Theorem 7.

**RQ2: Efficacy.** How does RPC improve reasoning performance compared to existing methods?

We evaluate the performance of PC and RPC in Figure 2 across various sample budgets. The results demonstrate that RPC achieves better performance than both PPL (which relies on internal LLM probabilities) and SC (which employs Monte Carlo sampling). The detailed accuracy results, including mean and standard deviation in Table 2 support these findings.

We also analyze the performance of PC separately. The results indicate that PC has a faster convergence rate than SC, which aligns with Theorem 4. The significant performance gains from PC to RPC, as shown in Figure 2a and Figure 2b, validate the effectiveness of the *Reasoning Pruning* module. This suggests that *Reasoning Pruning* helps reduce model errors when the LLM exhibits good alignment by eliminating incorrect reasoning paths that carry low LLM probability scores.

**RQ3: Reliability.** How does RPC enhance the reliability of confidence estimation compared to existing methods?

To evaluate the reliability of confidence estimation, we analyze the ECE results of RPC and comparison methods in Table 2. ECE measures the difference between predicted probabilities and empirical accuracy, as directly computing estimation error using ground-truth probabilities is virtually impractical. The results demonstrate that our RPC approach achieves lower ECE values and higher accuracy compared to PPL and SC, indicating more reliable confidence estimation. We visualize this improvement through reliability diagrams comparing SC and RPC in Figure 3 on MathOdyssey, which clearly shows the reduced calibration error of RPC.

### 4.3 Further Discussion

**Performance on Code Generation Tasks.** To investigate whether our proposed approaches can generalize to other reasoning tasks, such as code generation tasks, we evaluate RPC and comparison

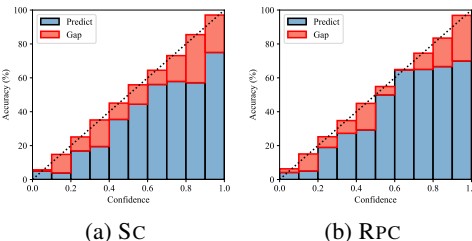

(a) Sc       (b) RPC

Figure 3: The reliability diagrams of Sc and RPC on MathOdyssey dataset using InternLM-2-MATH-Plus 7B model.

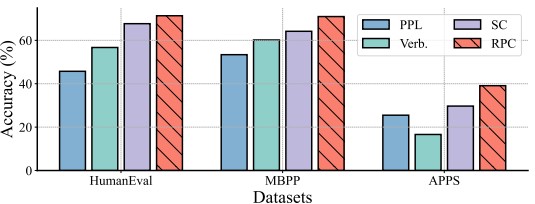

Figure 4: Performance on three code generation tasks using Deepseek-Coder 33B model. The experimental results show that our RPC achieves the best performance.

methods on three code generation benchmarks, as illustrated in Figure 4. The results show that RPC achieves the highest accuracy across all datasets, demonstrating its effectiveness beyond mathematics.

**Performance on Additional Reasoning Tasks.** To further validate the effectiveness of RPC, we conducted experiments on the GPQA [47] and LogiQA [39] benchmarks using the DeepSeek-R1-Distill-Qwen-7B model [21] with 16 samples. The results in Appendix D.5 are consistent with those from both math reasoning and code generation tasks, where RPC outperforms existing methods.

**Performance across Model Scales and Architectures.** To evaluate the generalization ability of our approaches across different model scales and architectures, we conducted additional experiments using InternLM2-Math-Plus 1.8B and DeepSeek-Math 7B models. The results in Table 3 is consistent with results in Table 2. We additionally report the detailed performance of InternLM2-Math-Plus 1.8B using diverse sampling budgets in Appendix D.2. The experimental results demonstrate the effectiveness of RPC across different model scales and architectures.

**Performance using Advanced Methods and Models.** To further verify the effectiveness of RPC across diverse scenarios, we conduct additional experiments using the DeepSeek-R1-Distill-Qwen-7B model [21], which is an advanced model with thinking capability. The results in Appendix D.4 demonstrate that the performance gains of RPC persist when combined with advanced models that have strong reasoning performance, highlighting its potential for enhancing LLM reasoning capability. Moreover, we also combine RPC with two advanced methods: ESC [36] and BoN using the reward model [69], which are advanced versions of Sc and PPL, respectively. The results in Appendix D.4 show that RPC applied to both methods consistently outperforms the Sc and their original versions, demonstrating its strong compatibility for integration with more advanced methods.

**Performance using High Sampling Temperatures.** As discussed above, sampling more diverse reasoning paths is important for improving reasoning performance. Therefore, we additionally conduct experiments under high sampling temperatures. As shown in Appendix D.1, RPC can further improve reasoning performance under high sampling temperatures by benefiting from the diverse reasoning paths, while the Sc may deteriorate due to increased estimation errors at high temperatures.

**Discussion about Computational Overhead.** We discuss the computational overhead of RPC in Appendix D.7. Theoretically, RPC introduces only minimal computational overhead compared to Sc. In practice, the additional computational overhead of RPC is negligible compared to the primary computational bottleneck (i.e., LLM inference time) in reasoning tasks. RPC actually provides an excellent computational trade-off, where the minimal computational overhead is exchanged for significant time savings achieved by reducing the number of required LLM inferences.

**Discussion about Hyper-parameters.** We provide a detailed discussion of the hyperparameters in Appendix D.6. Overall, RPC introduces the RP module to automatically prune reasoning paths with low probability without requiring manual threshold setting. This makes RPC a hyperparameter-free method that is robust across diverse reasoning tasks.

## 5 Related Work

This paper is related to the two topics, i.e., LLM Reasoning Boosting and LLM Confidence Estimation.

**LLM Reasoning Boosting.** Recent research [62, 38] has developed various methods to enhance LLM reasoning. CoT [31] proposes the "Let's think step by step" prompt to guide LLMs in generating

Table 3: Performance Comparison of different models and different parameter scales. The accuracy is reported as the mean and stdev. The best performance is highlighted in **bold**. The results show that our RPC outperforms existing methods in major cases.

| Method | InternLM2-Math-Plus 1.8B | | | | DeepSeekMath-RL 7B | | | |
|---|---|---|---|---|---|---|---|---|
| | MATH | MathOdyssey | OlympiadBench | AIME | MATH | MathOdyssey | OlympiadBench | AIME |
| PPL | $33.24 \pm 0.24$ | $\mathbf{16.56 \pm 0.88}$ | $3.08 \pm 0.20$ | $1.66 \pm 0.15$ | $42.51 \pm 0.23$ | $22.34 \pm 1.00$ | $5.90 \pm 0.31$ | $3.37 \pm 0.46$ |
| VERB | $7.21 \pm 0.17$ | $2.81 \pm 0.26$ | $0.77 \pm 0.06$ | $0.26 \pm 0.05$ | $14.29 \pm 0.23$ | $2.55 \pm 0.24$ | $2.36 \pm 0.15$ | $1.91 \pm 0.12$ |
| SC | $36.48 \pm 0.15$ | $14.52 \pm 0.46$ | $5.99 \pm 0.17$ | $2.66 \pm 0.20$ | $53.33 \pm 0.09$ | $36.68 \pm 0.65$ | $11.29 \pm 0.17$ | $9.42 \pm 0.23$ |
| RPC | $\mathbf{37.88 \pm 0.16}$ | $16.35 \pm 0.61$ | $\mathbf{6.52 \pm 0.24}$ | $\mathbf{3.26 \pm 0.24}$ | $\mathbf{53.37 \pm 0.11}$ | $\mathbf{37.25 \pm 0.69}$ | $\mathbf{11.30 \pm 0.11}$ | $\mathbf{9.52 \pm 0.31}$ |

structured solutions. For complex problems, Least-to-Most [72] introduces a decomposition strategy that breaks down challenges into manageable sub-problems. Few-shot methods [58, 17, 70] leverage carefully selected examples to improve reasoning performance. To enable more comprehensive reasoning, search-based methods [19] integrate Monte Carlo Tree Search (MCTS). Recent advancements have further enhanced MCTS by incorporating reward models [68, 46]. To directly optimize reasoning abilities, researchers have explored fine-tuning approaches [66, 34, 38, 65] using specialized datasets and reinforcement learning techniques [49, 41]. While these methods focus on generating accurate reasoning paths, our RPC can build upon them by utilizing multiple sampling strategies, enabling better reasoning performance. Recent studies recognize its computational issues and propose early stopping [37] and dynamic sampling [56, 54, 2] to improve efficiency. While previous studies primarily focus on developing specific methods that demonstrate empirical effectiveness, our paper introduces a theoretical framework for analyzing them and provides theoretical insights.

**LLM Confidence Estimation.** The confidence estimation for LLM can be categorized into three types: (1) perplexity confidence, (2) verbalized confidence, and (3) self-consistency confidence. Perplexity confidence [28, 15] utilizes the geometric mean of LLM prediction probabilities (i.e., perplexity [10, 6]) to evaluate model adherence [44] and prompt quality [64]. Verbalized confidence [30, 60, 52] directly asks LLMs to express their confidence through various approaches, such as multi-agent deliberation [61], multi-step evaluation [60], top-k ranking [52], few-shot prompting [40], and reflection [14, 71]. Self-consistency confidence [57, 11, 12, 26, 50] measures the agreement among multiple generated answers to improve reasoning performance, with recent work [60, 1, 4] further developing this approach as a confidence metric. Recent studies have identified that LLM confidence is crucial for enhancing LLM reasoning, such as DeepConf [18], CISC [51], and TTSC [27], which align with our core idea. Our work provides theoretical explanations to support their findings.

# 6 Conclusion

In this paper, we present a theoretical framework for LLM reasoning problem in sampling-based test-time scaling, providing insights and principled guidance for designing better LLM reasoning methods. This framework decomposes LLM reasoning error into estimation error and model error, revealing the efficiency issues of the self-consistency method and effectiveness and degeneration issues of the perplexity method. To this end, we introduce ***Reasoning-pruning Perplexity Consistency*** (RPC), which integrates internal LLM probabilities into the self-consistency framework for faster estimation error convergence while maintaining low model error and prunes low-probability reasoning paths to address the degeneration issue. Both theoretical analysis and empirical results demonstrate that RPC achieves superior error convergence, reasoning performance, and confidence reliability.

**Limitations and Future Work.** Despite our theoretical analysis providing insights and guidance for designing better LLM reasoning methods, this paper presents only a preliminary step toward building advanced reasoning approaches. We believe our theoretical framework can guide future research in this promising direction. Detailed limitations and future works are discussed in Appendix E.

## Broader Impacts

This paper presents a new theoretical framework for analyzing sampling-based test-time scaling methods in LLM reasoning, enabling researchers to identify the limitations of existing methods and to provide insights for developing future approaches. Therefore, we believe that this work has the potential to benefit the broader LLM reasoning community by offering a theoretical foundation. As this paper is a theoretical study, we do not identify negative societal impacts that need to be discussed.

## Acknowledgments and Disclosure of Funding

We appreciate the anonymous reviewers for their valuable insights and helpful comments. This work is supported by the National Natural Science Foundation of China (Grant No. 624B2068 and 62576162), the Leading-edge Technology Program of Jiangsu Science Foundation (BK20232003), the Key Program of Jiangsu Science Foundation (BK20243012), and the Fundamental Research Funds for the Central Universities (022114380023).

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

# A  Theoretical Results

## A.1  Proof of Proposition 1, Proposition 2 and Proposition 3

*Proof.* In this proof, all the expectations are taken over the combinations of $n$-sampled reasoning paths $\tilde{t}_1, \ldots, \tilde{t}_n$ and we omit the details of expectation for simplicity. We first prove the general error decomposition in Proposition 1. Then, we give the proofs of Proposition 2 and Proposition 3. We adopt a common and practical assumption that the LLM sampling can be modeled as sampling from a Bernoulli distribution [55], which allows us to further compute the estimation error for specific methods.

**Proof of General Error Decomposition**  For any input $x$ with ground-truth answer $y$ and any possible answer $\hat{y}$, let $\hat{p}(\hat{y} \mid x)$ denote the unbiased estimated confidence of $\hat{y}$ and $p(\hat{y} \mid x)$ denote the ground truth confidence. We have

$$
\begin{aligned}
\mathcal{E}_{\hat{p}}(\hat{y}) &= \mathbb{E}\left[\left(\hat{p}(\hat{y} \mid x) - \mathbb{I}[\hat{y} = y]\right)^2\right] \\
&= \mathbb{E}\left[\left(\hat{p}(\hat{y} \mid x) - p(\hat{y} \mid x) + p(\hat{y} \mid x) - \mathbb{I}[\hat{y} = y]\right)^2\right] \\
&= \underbrace{\mathbb{E}\left[\left(\hat{p}(\hat{y} \mid x) - p(\hat{y} \mid x)\right)^2\right]}_{\text{Estimation Error}} + \underbrace{\left(p(\hat{y} \mid x) - \mathbb{I}[\hat{y} = y]\right)^2}_{\text{Model Error}} \\
&\quad + \underbrace{2\mathbb{E}\left[\left(\hat{p}(\hat{y} \mid x) - p(\hat{y} \mid x)\right)\left(p(\hat{y} \mid x) - \mathbb{I}[\hat{y} = y]\right)\right]}_{\text{Cross Term}}
\end{aligned}
\tag{11}
$$

Then, we prove the Cross Term is zero for the unbiased confidence function as follows:

$$
\begin{aligned}
\text{Cross Term} &= 2\left(p(\hat{y} \mid x) - \mathbb{I}[\hat{y} = y]\right) \cdot \mathbb{E}\left[\hat{p}(\hat{y} \mid x) - p(\hat{y} \mid x)\right] \\
&= 2\left(p(\hat{y} \mid x) - \mathbb{I}[\hat{y} = y]\right) \cdot 0 \\
&= 0
\end{aligned}
\tag{12}
$$

Therefore, we have the general error decomposition:

$$
\mathcal{E}_{\hat{p}}(\hat{y}) = \underbrace{\mathbb{E}\left[\left(\hat{p}(\hat{y} \mid x) - p(\hat{y} \mid x)\right)^2\right]}_{\text{Estimation Error}} + \underbrace{\left(p(\hat{y} \mid x) - \mathbb{I}[\hat{y} = y]\right)^2}_{\text{Model Error}}
\tag{13}
$$

**Proof of Error Decomposition for Sc**  For any input $x$ with ground-truth answer $y$, let $\hat{p}^{(\text{SC})}(\hat{y} \mid x)$ denote the estimated probability of $\hat{y}$ by Sc method for any possible answer $\hat{y}$. The confidence function of Sc is $\hat{p}^{(\text{SC})}(\hat{y} \mid x) = \frac{1}{n}\sum_{i=1}^{n}\mathbb{I}[\tilde{y}_i = \hat{y}_i]$, where $\tilde{y}_1 = g(\tilde{t}_1), \ldots, \tilde{y}_n = g(\tilde{t}_n)$ and $\tilde{t}_1, \ldots, \tilde{t}_n$ are $n-$sampled reasoning paths from the LLM distribution. Apply the error decomposition, we have

$$
\begin{aligned}
\mathcal{E}_{\hat{p}^{(\text{SC})}}(\hat{y}) &= \mathbb{E}\left[\left(\hat{p}^{(\text{SC})}(\hat{y} \mid x) - p(\hat{y} \mid x)\right)^2\right] + \left(p(\hat{y} \mid x) - \mathbb{I}[\hat{y} = y]\right)^2 \\
&= \frac{1}{n}p(\hat{y} \mid x)(1 - p(\hat{y} \mid x)) + \left(p(\hat{y} \mid x) - \mathbb{I}[\hat{y} = y]\right)^2.
\end{aligned}
\tag{14}
$$

**Proof of Error Decomposition for Ppl**  Another way is to use the confidence function to build the sampling probability distribution using the unique set of $n$ sampled reasoning paths $\mathcal{R} = \text{Set}\left(\tilde{t}_1, \ldots, \tilde{t}_n\right)$, i.e.,

$$
\hat{p}^{(\text{PPL})}(\hat{t} \mid x) = \begin{cases} p(\tilde{t}_i \mid x), & \text{if there is } \tilde{t}_i = \hat{t} \\ 0, & \text{otherwise} \end{cases} = \sum_{\tilde{t} \in \mathcal{R}}\mathbb{I}(\tilde{t} = \hat{t})p(\tilde{t} \mid x).
\tag{15}
$$

To simplify the analysis, we assume that each $\tilde{t}_1, \ldots, \tilde{t}_n$ is different from each other, which is a reasonable assumption in practical LLM reasoning and slightly affects the value of $n$ if this assumption is not satisfied. Then, we have

$$
\begin{aligned}
\mathbb{E}\left[\hat{p}^{(\text{PPL})}(\hat{t} \mid x) - p(\hat{t} \mid x)\right] &= -(1 - p(\hat{t} \mid x))^n p(\hat{t} \mid x) \\
\mathbb{E}\left[(\hat{p}^{(\text{PPL})}(\hat{t} \mid x) - p(\hat{t} \mid x))^2\right] &= (1 - p(\hat{t} \mid x))^n p(\hat{t} \mid x)^2.
\end{aligned}
\tag{16}
$$

Hence,

$$
\begin{aligned}
\mathcal{E}_{\hat{p}^{(\text{PPL})}}(\hat{t}) &= \mathbb{E}\big[(\hat{p}^{(\text{PPL})}(\hat{t}\,|\,x) - \mathbb{I}[\hat{y}=y])^2\big] \\
&= \mathbb{E}\big[(\hat{p}^{(\text{PPL})}(\hat{t}\,|\,x) - p(\hat{t}\,|\,x) + p(\hat{t}\,|\,x) - \mathbb{I}[g(\hat{t})=y])^2\big] \\
&= -(1 - p(\hat{t}\,|\,x))^n p(\hat{t}\,|\,x)^2 + 2(1 - p(\hat{t}\,|\,x))^n p(\hat{t}\,|\,x)\mathbb{I}[g(\hat{t})=y] + (p(\hat{t}\,|\,x) - \mathbb{I}[g(\hat{t})=y])^2 \\
&= (1 - p(\hat{t}\,|\,x))^n p(\hat{t}\,|\,x)(2\mathbb{I}[g(\hat{t})=y] - p(\hat{t}\,|\,x)) + (p(\hat{t}\,|\,x) - \mathbb{I}[g(\hat{t})=y])^2.
\end{aligned}
\tag{17}
$$

Hence, we complete the proof. $\qquad\square$

## A.2 Model Error Comparison in Ideal Case

**Proposition 4** (Comparison of Model Errors). *Consider a setting with infinite sampling of reasoning paths ($n \to \infty$) where each incorrect reasoning path leads to a unique answer, that is, $g(\tilde{t}_i) \neq g(\tilde{t}_j)$ for any $i \neq j$ where $g(\tilde{t}_i) \neq y$ and $g(\tilde{t}_j) \neq y$. The model error of self-consistency ($\mathcal{E}_{Model}^{(\text{SC})}$) and perplexity ($\mathcal{E}_{Model}^{(\text{PPL})}$) satisfy:*

$$
\mathcal{E}_{Model}^{(\text{SC})} \leq \mathcal{E}_{Model}^{(\text{PPL})}
\tag{18}
$$

*The inequality is strict when the consistency function identifies equivalent correct reasoning paths more than once.*

*Remark* 9. Although the assumptions in Proposition 4 are idealized, this special case demonstrates that the consistency function in self-consistency achieves better model error than the perplexity method. In practice, the perplexity method always gives larger model error compared to the self-consistency method, as it does not leverage the consistency function to analyze the structural properties of specific reasoning problems. The proof is presented as follows.

*Proof.* We first recall the definitions of the model error for self-consistency and perplexity:

$$
\begin{cases}
\mathcal{E}_{\text{Model}}^{(\text{SC})} &= \sum_{\hat{y} \in \{g(\tilde{t}_i)\,|\,i=1\ldots n\}} \big(\hat{p}^{(\text{SC})}(\hat{y}\,|\,x) - \mathbb{I}[\hat{y}=y]\big)^2, \\
\mathcal{E}_{\text{Model}}^{(\text{PPL})} &= \sum_{\hat{t} \in \mathcal{R}} \big(\hat{p}^{(\text{PPL})}(\hat{t}\,|\,x) - \mathbb{I}[g(\hat{t})=y]\big)^2.
\end{cases}
\tag{19}
$$

where $\mathcal{R} = \text{Set}(\tilde{t}_1, \ldots, \tilde{t}_n)$ is the set of reasoning paths sampled from the LLM. We can compute the difference between the model error of SC and PPL as follows:

$$
\begin{aligned}
\mathcal{E}_{\text{Model}}^{(\text{SC})} - \mathcal{E}_{\text{Model}}^{(\text{PPL})} &= \sum_{\hat{y} \in \{g(\tilde{t}_i)\,|\,i=1\ldots n\}} \big(\hat{p}^{(\text{SC})}(\hat{y}\,|\,x) - \mathbb{I}[\hat{y}=y]\big)^2 - \sum_{\hat{t} \in \mathcal{R}} \big(\hat{p}^{(\text{PPL})}(\hat{t}\,|\,x) - \mathbb{I}[g(\hat{t})=y]\big)^2 \\
&= \sum_{\hat{y} \in \{g(\hat{t}_i)\,|\,i=1\ldots n\}} \underbrace{\Big[\big(\hat{p}^{(\text{SC})}(\hat{y}\,|\,x) - \mathbb{I}[\hat{y}=y]\big)^2 - \sum_{\hat{t} \in \mathcal{R}} \mathbb{I}[g(\hat{t})=\hat{y}]\big(\hat{p}^{(\text{PPL})}(\hat{t}\,|\,x) - \mathbb{I}[\hat{y}=y]\big)^2\Big]}_{(A)}
\end{aligned}
\tag{20}
$$

Assuming infinite samplings, the unbiasedness of SC gives us:

$$
\hat{p}^{(\text{SC})}(\hat{y}\,|\,x) = \frac{1}{n}\sum_{i=1}^{n} \mathbb{I}[g(\tilde{t}_i)=\hat{y}] = \sum_{\hat{t} \in \mathcal{R}} \mathbb{I}[g(\hat{t})=\hat{y}] \cdot \hat{p}^{(\text{PPL})}(\hat{t}\,|\,x).
\tag{21}
$$

For any $\hat{y}$, consider two cases:

(i) If $\hat{y} = y$, then $\hat{y}$ is the correct answer. We have

$$(A) = \left(\hat{p}^{(\text{Sc})}(\hat{y} \mid x) - 1\right)^2 - \sum_{\hat{t} \in \mathcal{R}} \mathbb{I}[g(\hat{t}) = \hat{y}]\left(\hat{p}^{(\text{Ppl})}(\hat{t} \mid x) - 1\right)^2$$

$$= \left(\hat{p}^{(\text{Sc})}(\hat{y} \mid x)^2 + 1 - 2\hat{p}^{(\text{Sc})}(\hat{y} \mid x)\right) - \sum_{\hat{t} \in \mathcal{R}} \mathbb{I}[g(\hat{t}) = \hat{y}]\left(\hat{p}^{(\text{Ppl})}(\hat{t} \mid x)^2 + 1 - 2\hat{p}^{(\text{Ppl})}(\hat{t} \mid x)\right)$$

$$= \hat{p}^{(\text{Sc})}(\hat{y} \mid x)^2 + 1 - \sum_{\hat{t} \in \mathcal{R}} \mathbb{I}[g(\hat{t}) = \hat{y}] \cdot \hat{p}^{(\text{Ppl})}(\hat{t} \mid x)^2 - \sum_{\hat{t} \in \mathcal{R}} \mathbb{I}[g(\hat{t}) = \hat{y}]$$

$$\leq \hat{p}^{(\text{Sc})}(\hat{y} \mid x)^2 + 1 - \frac{\hat{p}^{(\text{Sc})}(\hat{y} \mid x)^2}{\sum_{\hat{t} \in \mathcal{R}} \mathbb{I}[g(\hat{t}) = \hat{y}]} - \sum_{\hat{t} \in \mathcal{R}} \mathbb{I}[g(\hat{t}) = \hat{y}]$$

$$\tag{22}$$

Let $\hat{p}^{(\text{Sc})}(\hat{y} \mid x)^2 = B^2$ and $\sum_{\hat{t} \in \mathcal{R}} \mathbb{I}[g(\hat{t}) = \hat{y}] = C$, then

$$\begin{aligned}
(A) &\leq B^2 + 1 - \frac{B^2}{C} - C \\
&= \frac{1}{C}\left[B^2 C + C - B^2 - C^2\right] \\
&= \frac{1}{C}\left[(C - B^2)(1 - C)\right] \\
&\leq 0.
\end{aligned} \tag{23}$$

Equality holds if $\sum_{\hat{t} \in \mathcal{R}} \mathbb{I}[g(\hat{t}) = \hat{y}] = C = 1$. This indicates that $(A) < 0$ if the consistency function is effective at least once, making $\sum_{\hat{t} \in \mathcal{R}} \mathbb{I}[g(\hat{t}) = \hat{y}] = C > 1$.

(ii) If $\hat{y} \neq y$, then $\hat{y}$ is incorrect. Assuming distinct answers for wrong reasoning paths, we have $\sum_{\hat{t} \in \mathcal{R}} \mathbb{I}[g(\hat{t}) = \hat{y}] = 1$, thus

$$\begin{aligned}
(A) &= \left[\left(\hat{p}^{(\text{Sc})}(\hat{y} \mid x) - \mathbb{I}[\hat{y} = y]\right)^2 - \sum_{\hat{t} \in \mathcal{R}} \mathbb{I}[g(\hat{t}) = \hat{y}]\left(\hat{p}^{(\text{Ppl})}(\hat{t} \mid x) - \mathbb{I}[\hat{y} = y]\right)^2\right] \\
&= \left[\left(\hat{p}^{(\text{Sc})}(\hat{y} \mid x) - \mathbb{I}[\hat{y} = y]\right)^2 - \left(\hat{p}^{(\text{Sc})}(\hat{y} \mid x) - \mathbb{I}[\hat{y} = y]\right)^2\right] = 0,
\end{aligned} \tag{24}$$

since only one $\hat{t} \in \mathcal{R}$ satisfying that $g(\hat{t})$ equals the incorrect answer $\hat{y}$.

Therefore, $\mathcal{E}_{\text{Model}}^{(\text{Ppl})} - \mathcal{E}_{\text{Model}}^{(\text{Sc})} \leq 0$, indicating that the model error of self-consistency is always less than or equal to the model error of perplexity under our assumptions. Moreover, if the consistency function is effective at least once, the model error of self-consistency is strictly less than the model error of perplexity. $\square$

## A.3 Proof of Theorem 4

*Proof.* In this proof, all the expectations are taken over the combinations of $n$-sampled reasoning paths $\tilde{t}_1, \ldots, \tilde{t}_n$ for estimating the confidence $\hat{p}^{(\text{Pc})}$ and we omit the details of expectation for simplicity. To simplify the analysis, we assume that each $\tilde{t}_1, \ldots, \tilde{t}_n$ is different from each other, which is a reasonable assumption in practical LLM reasoning and slightly affects the value of $n$ if this assumption is not satisfied. For any given answer $\hat{y}$, the estimated probability of Pc is $\hat{p}(\hat{y} \mid x) = \sum_{i=1}^{n} \mathbb{I}[g(\tilde{t}_i) = \hat{y}]p(\tilde{t}_i \mid x)$, allowing the reasoning error of Pc can be computed as follows.

$$\begin{aligned}
\mathcal{E}_{\hat{p}^{(\text{Pc})}}(\hat{y}) &= \mathbb{E}\left[(\hat{p}^{(\text{Pc})}(\hat{y} \mid x) - \mathbb{I}[\hat{y} = y])^2\right] \\
&= \mathbb{E}\left[(\hat{p}^{(\text{Pc})}(\hat{y} \mid x) - p(\hat{y} \mid x) + p(\hat{y} \mid x) - \mathbb{I}[\hat{y} = y])^2\right].
\end{aligned} \tag{25}$$

We define $k := |\{\tilde{t} \mid g(\tilde{t}) = \hat{y}\}|$, which means that how many reasoning paths whose answers are $\hat{y}$ can be covered given limited sampling budget. Note that we further have $\mathbb{E}[\mathbb{I}[g(\tilde{t}) = \hat{y}]p(\tilde{t} \mid x)] =$

$\frac{1}{k}p(\hat{y} \mid x)$, thus

$$\mathbb{E}[\hat{p}^{(\text{PC})}(\hat{y} \mid x)] = \mathbb{E}\Big[\sum_{i=1}^{n} \mathbb{I}[g(\tilde{t}_i) = \hat{y}]p(\tilde{t}_i \mid x)\Big]$$
$$= \Big(1 - \big(1 - \frac{1}{k}p(\hat{y} \mid x)\big)^n\Big)p(\hat{y} \mid x) \tag{26}$$
$$= (1 - \alpha^n)p(\hat{y} \mid x),$$

where $\alpha := \frac{1}{k}p(\hat{y} \mid x)$. Again, we have

$$\mathbb{E}[\hat{p}^{(\text{PC})}(\hat{y} \mid x) - p(\hat{y} \mid x)] = -\big(1 - \alpha\big)^n p(\hat{y} \mid x))$$
$$\mathbb{E}[(\hat{p}^{(\text{PC})}(\hat{y} \mid x) - p(\hat{y} \mid x))^2] = \Big(1 - \big(1 - \alpha\big)^n\Big)\big(1 - \alpha\big)^n p(\hat{y} \mid x))^2 \tag{27}$$

Hence,

$$\mathcal{E}_{\hat{p}^{(\text{PC})}}(\hat{y}) = \mathbb{E}\big[(\hat{p}^{(\text{PC})}(\hat{y} \mid x) - \mathbb{I}[\hat{y} = y])^2\big]$$
$$= \mathbb{E}\big[(\hat{p}^{(\text{PC})}(\hat{y} \mid x) - p(\hat{y} \mid x) + p(\hat{y} \mid x) - \mathbb{I}[\hat{y} = y])^2\big]$$
$$= (1 - \alpha)^n p(\hat{y} \mid x)\big(2\mathbb{I}[\hat{y} = y] - (1 + (1 - \alpha)^n)p(\hat{y} \mid x)\big) + (p(\hat{y} \mid x) - \mathbb{I}[\hat{y}_i = y])^2, \tag{28}$$

which completes the proof. $\qquad\square$

### A.4 Proof of Theroem 7

*Proof.* Let us set the pruning threshold by $\tau := p(y \mid x)$. Then, we have

$$\mathcal{E}_{\hat{p}^{(\text{RPC})}}(\hat{y}) = \underbrace{\alpha p(\hat{y} \mid x)\big(2\mathbb{I}[\hat{y} = y] - (1 + \alpha)p(\hat{y} \mid x)\big)\mathbb{I}[(p(\hat{y}) \mid x) < \tau]}_{\text{Estimation Error}}$$
$$+ \underbrace{(p(\hat{y} \mid x) - \mathbb{I}[\hat{y}_i = y])^2 \mathbb{I}[(p(\hat{y}) \mid x) < \tau]}_{\text{Model Error}} \tag{29}$$

However, we only have an estimation of $p(\hat{y} \mid x)$, i.e., $\hat{p}^{(\text{RPC})}(\hat{y} \mid x) = k\mathbb{E}\big[\mathbb{I}[g(\tilde{t}) = \hat{y}]p(\tilde{t} \mid x)\big] \approx \frac{k}{\hat{k}}\sum_{i=1}^{k} p(\tilde{t}_i \mid x)$, where $\tilde{t}_1, \ldots, \tilde{t}_{\hat{k}}$ are $\hat{k}$ sampled reasoning paths whose answer is $\hat{y}$. Therefore, the reasoning error of our approximate version can be computed by

$$\mathcal{E}_{\hat{p}^{(\text{RPC})}}(\hat{y}) = \underbrace{\alpha(p)p(\hat{y} \mid x)\big(2\mathbb{I}[\hat{y} = y] - (1 + \alpha(p))p(\hat{y} \mid x)\big)\mathbb{I}[\frac{1}{k}\sum_{i=1}^{\hat{k}} p(\tilde{t}_i \mid x) < \frac{1}{m}\tau]}_{\text{Estimation Error}}$$
$$+ \underbrace{(p(\hat{y} \mid x) - \mathbb{I}[\hat{y}_i = y])^2 \mathbb{I}\big[\frac{1}{\hat{k}}\sum_{i=1}^{\hat{k}} p(\tilde{t}_i \mid x) < \frac{1}{m}\tau\big]}_{\text{Model Error}} \tag{30}$$

Hence, we only need to consider the probability $\frac{1}{\hat{k}}\sum_{i=1}^{\hat{k}} p(\tilde{t}_i \mid x) > \frac{1}{m}\tau$. Using Hoeffding's inequality, we can obtain that

$$\mathbb{P}\Big(\frac{1}{\hat{k}}\sum_{i=1}^{\hat{k}} p(\tilde{t}_i \mid x) - \frac{1}{k}p(\hat{y} \mid x) \geq \tau\Big) \leq \exp\Big(-\frac{2\hat{k}\gamma^2}{p(\hat{y} \mid x)^2}\Big) \tag{31}$$

We set $\gamma = \tau - \frac{1}{k}p(\hat{y} \mid x) = \tau + \alpha - 1$, then

$$\mathbb{P}\Big(\frac{1}{\hat{k}}\sum_{i=1}^{\hat{k}} p(\tilde{t}_i \mid x) \geq \tau\Big) \leq \exp\Big(-2\hat{k}k^2(1 - \frac{\tau}{1 - \alpha})^2\Big). \tag{32}$$

Hence, we complete the proof. $\qquad\square$

# B    Pseudo Code of RPC Method

In this section, we provide the pseudo code of RPC. The output of Algorithm 1 is a set of reasoning paths with the highest confidence. The extraction function $g(\cdot)$ can be used to parse the reasoning paths to answers.

---

**Algorithm 1** Reasoning-pruning Perplexity Consistency

---

**Input:**
 1: Sampled Reasoning paths $\tilde{t}_1, \ldots, \tilde{t}_n$
 2: LLM Internal Probabilities $p_1, \ldots, p_n$
 3: Consisitency function $\mathbb{I}_C(\cdot, \cdot)$
**Output:** Most-confident reasoning paths with probabilities
 4: {**Phase 1: Reasoning Pruning**}
 5: $(k_1, \lambda_1, k_2, \lambda_2, w_1, w_2) \leftarrow$ Model probability distribution parameters from $p_1, \ldots, p_n$ {Using Theorem 3.2}
 6: $p_{\text{mean}} \leftarrow \frac{1}{n} \sum_{i=1}^{n} p_i$
 7: $\text{I}_{\text{retain}} \leftarrow \{i \mid P_{\text{High}}(p_i) > 0.5 \text{ or } p_i \geq p_{\text{mean}}\}$ {$P_{\text{High}}$ is defined in Theorem 3.2}
 8: {**Phase 2: Perplexity Consistency**}
 9: $\text{U} \leftarrow \text{Set}(\tilde{t}_1, \ldots, \tilde{t}_n)$
10: $\text{I}_{\text{unique}} \leftarrow \{i \mid \tilde{t}_i \in \text{U} \text{ and } i \in \text{I}_{\text{retain}}\}$
11: **for** each reasoning path $\tilde{t} \in \text{U}$ **do**
12:     $\text{C}(\tilde{t}) \leftarrow \sum_{i \in \mathcal{I}_{\text{retain}}} \mathbb{I}_C[\tilde{t}, \tilde{t}_i] p_i$
13: **end for**
14: $\text{C}_{\text{max}} \leftarrow \max_{\tilde{t} \in \text{U}} \text{C}(\tilde{t})$
15: **return** $\{(\tilde{t}, \text{C}(\tilde{t})) \mid \tilde{t} \in \text{U}, \text{C}(\tilde{t}) = \text{C}_{\text{max}}\}$

---

# C    Detailed Experimental Settings

## C.1    Datasets

For mathematical reasoning tasks, we evaluate our proposed methods and comparison methods on four mathematical datasets that include MATH, MathOdyssey, OlympiadBench, and AIME datasets.

- MATH dataset [25] is a dataset comprised of challenging competition math problems and we use its 5,000 testing data for evaluation.
- MathOdyssey dataset [16] contains 387 problems, covering advanced high-school level, university-level, and Olympiad-level mathematics.
- OlympiadBench dataset [23] contains 8,476 Olympiad-level mathematics and physics problems. We select the English problems without images, resulting in a testing dataset of 1,284 problems.
- AIME dataset [67] contains 993 test problems collected from the American Invitational Mathematics Examination, spanning from 1983 to 2024.

For code generation tasks, we conduct experiments on three common benchmark datasets. HumanEval [8] contains 164 hand-written Python programming problems. MBPP [3](sanitized version) consists of 427 entry-level programming problems. We also include the introductory-level problems of APPS [24], which contains 1000 problems.

## C.2    Detailes of Mathematical Reasoning Task

For all experiments in the main paper, we use a sampling temperature of 1.0 and set the top-p parameter to 0.95. The prompt template follows the recommendation of each LLM.

**Prompt for Math Reasoning Tasks.**    The InternLM2-MATH-Plus 1.8B and 7B models are chat models that facilitate conversations between two roles: "user" and "assistant". The prompt for the "user" role is provided in Prompt 1. In contrast, the DeepSeek-Math 7B model operates in a non-chat mode, and its corresponding prompt is shown in Prompt 2.

> **Prompt 1: Prompt for InternLM-2-Math-Plus**
>
> Problem:
> {instruction}
> Let's think step by step
> Solution:

> **Prompt 2: Prompt for DeepSeek-Math**
>
> {instruction}
> Please reason step by step, and put your final answer within \boxed{}.

**Prompt for Mathematical Verbalized Method.** We observed that the tuned math models are challenging to prompt for generating confidence. Therefore, we adopted the methods from [52] to calculate the probability based on the likelihood of the first generated "True" token and the first generated "False" token. The corresponding prompt is provided in Prompt 3.

> **Prompt 3: Prompt for Mathematical Verbalized Method**
>
> Question: {question}
> Proposed Answer: {answer}
> Is the proposed answer:
> \t(A) True or
> \t(B) False?
> The proposed answer is:

### C.3 Detailes of Code Generation Task

**Code Generation.** On the code generation task, we let LLM generate a code snippet to solve a given programming problem, and then evaluate its functional correctness based on the ground-truth test cases provided by the dataset. In detail, we set the top $p$ to 0.95, and the max generation length to 1024. For code snippet post-processing, we first extract the code text from the code block surrounded by triple-backticks("'''"), and then we follow [8] to truncate the generated code snippet before the following stop sequences: "\nclass", "\ndef", "\n#", "\nif", "\nprint". At the same time, we also obtain the log-probability of each token from the LLM response. For "verbalization" setting, the verbalized confidence is also extracted from the text generated by LLM along with the code snippet.

**Self-consistency on Code.** We follow [7] to sample 100 test cases for each programming problem from the same model. Then, we achieved self-consistency in code at the semantic equivalence level, which is based on the execution behavior of any two codes on this set of test cases. More formally, we implemented the consistency function $\mathbb{I}_C(\cdot, \cdot)$ as an indicator function that indicates whether two codes are semantically equivalent, i.e., $\mathbb{I}_C(x, y) = 1$ if and only if code $x$ and $y$ execute the same result on this set of test cases.

**Prompt for Generating Code.** The prompt for generating code consists of a header, a functional signature, and a docstring and LLM needs to implement the body of this function. An Illustration of `has_close_elements` problem is shown in Prompt 4.

**Prompt for Generating Test Cases.** For generating test cases, we implement the function body with a "pass" statement on the basis of the prompt to generate the code, and added a comment to require the LLM to generate test cases for the programming problem. An Illustration of `has_close_elements` problem is shown in Prompt 5.

```python
def has_close_elements(numbers: List[float],
    threshold: float) -> bool:
"""Check if in given list of numbers, are any two numbers
closer to each other than given threshold.
"""
```

```python
def has_close_elements(numbers: List[float],
    threshold: float) -> bool:
""" Check if in given list of numbers, are any two numbers
closer to each other than given threshold.
"""
    pass
# check the correctness of has_close_elements
assert
```

**Prompt for Code Verbalized Method.**   For generating code with verbalized confidence, we add instructions for generating verbalized confidence, as well as format requirements to facilitate the extraction of code and confidence score. We also give a simple example to help LLM understand the format requirements at the end of the prompt. An Illustration of `has_close_elements` problem is shown in Prompt 6.

Come up with a solution that solves the following programming question and provide your confidence score in this solution like 0%, 10%, ... 100%.

```python
def has_close_elements(numbers: List[float],
    threshold: float) -> bool:
"""Check if in given list of numbers, are any two numbers
closer to each other than given threshold.
"""
```

Format requirement: output in the form of the following example. Do not provide any additional explanations. Here is an output example:
Solution:

```python
your code ...
```

Confidence:

# D   Detailed Experimental Results

## D.1   Results under High Sampling Temperature.

Using a high sampling temperature enables LLMs to produce more diverse outputs, potentially enhancing reasoning performance. However, it also leads to an increase in estimation error.

To investigate the effectiveness of our approaches in addressing the estimation error issue, we conducted experiments with higher sampling temperatures (T = 1.1 and T = 1.3) using the InternLM-2-MATH-Plus 7B model. The results in Table 4 indicate that our RPC approach consistently surpasses baseline methods. Notably, a significant performance gap persists between RPC and SC, indicating that RPC effectively tackles the estimation error issue even under sampling high-temperature.

We also report the performance of RPC with different sampling numbers under the high temperature in Figure 5 and Figure 6. The results demonstrate that the RPC method effectively improves the reasoning performance at higher temperatures, as it leverages the increased diversity in sampling to enhance self-consistency. In contrast, the SC method's performance deteriorates due to increased estimation errors at higher temperatures.

Table 4: Performance Comparison of different models and different parameter scales. The accuracy is reported as the mean and stdev. The best performance is highlighted in **bold**. The results show that our RPC approach consistently outperforms existing methods.

| Method | Temperature = 1.1 | | | |
| --- | --- | --- | --- | --- |
| | MATH | MathOdyssey | OlympiadBench | AIME |
| PPL | $47.35 \pm 0.16$ | $28.59 \pm 1.30$ | $7.27 \pm 0.23$ | $6.02 \pm 0.34$ |
| VERB | $25.51 \pm 0.23$ | $9.41 \pm 0.44$ | $3.66 \pm 0.16$ | $3.07 \pm 0.15$ |
| SC | $50.66 \pm 0.22$ | $27.89 \pm 0.43$ | $10.74 \pm 0.15$ | $8.73 \pm 0.24$ |
| RPC | $\mathbf{52.58 \pm 0.14}$ | $\mathbf{32.98 \pm 0.69}$ | $\mathbf{11.00 \pm 0.24}$ | $\mathbf{9.30 \pm 0.29}$ |

| Method | Temperature = 1.3 | | | |
| --- | --- | --- | --- | --- |
| | MATH | MathOdyssey | OlympiadBench | AIME |
| PPL | $47.58 \pm 0.31$ | $26.38 \pm 1.41$ | $7.76 \pm 0.46$ | $6.50 \pm 0.41$ |
| VERB | $24.62 \pm 0.33$ | $8.60 \pm 0.26$ | $3.11 \pm 0.17$ | $2.29 \pm 0.12$ |
| SC | $50.65 \pm 0.14$ | $27.61 \pm 0.67$ | $10.49 \pm 0.18$ | $8.02 \pm 0.20$ |
| RPC | $\mathbf{53.12 \pm 0.14}$ | $\mathbf{33.19 \pm 0.56}$ | $\mathbf{10.91 \pm 0.18}$ | $\mathbf{8.83 \pm 0.23}$ |

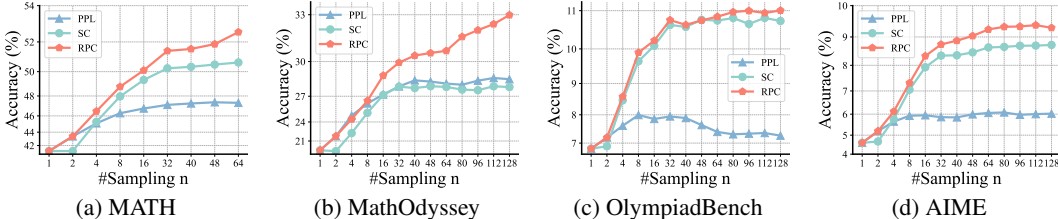

| (a) MATH | (b) MathOdyssey | (c) OlympiadBench | (d) AIME |

Figure 5: The accuracy of InternLM-2-MATH-Plus 7B model on four mathematical reasoning datasets with different sample sizes $n$. The sampling temperature is set to 1.1.

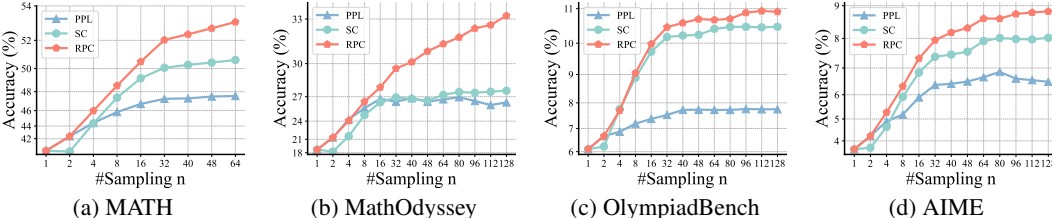

| (a) MATH | (b) MathOdyssey | (c) OlympiadBench | (d) AIME |

Figure 6: The accuracy of InternLM-2-MATH-Plus 7B model on four mathematical reasoning datasets with different sample sizes $n$. The sampling temperature is set to 1.3.

## D.2 Performance with Different Models Scales

In the Figure 2, we plot the accuracy of the InternLM-2-MATH-Plus 7B model on four mathematical reasoning datasets with different sample sizes $n$. Here, we further investigate the performance of relateively small model, InternLM-2-MATH-Plus 1.8B, is presented in Figure 7. Similar conclusions can be drawn from these results. For the MathOdyssey dataset, the PPL method shows superior performance compared to other methods, which can be attributed to the relatively low model error of PPL on this dataset, allowing the perplexity-based approach to function effectively. Furthermore, the RPC method consistently outperforms the SC method, which demonstrates its ability to enhance the convergence properties of the SC method.

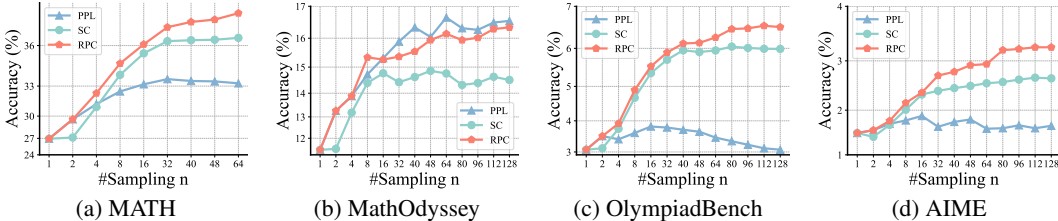

|  (a) MATH | (b) MathOdyssey | (c) OlympiadBench | (d) AIME |

Figure 7: The accuracy of InternLM-2-MATH-Plus 1.8B model on four mathematical reasoning datasets with different sample sizes $n$.

## D.3 Performance with R1 LLMs

We further investigate the performance of each method using R1 LLMs with thinking capability [9]. Here, we conduct experiments on the DeepSeek-R1-Distill-Qwen-7B model [21] using 16 samples, where the internal probability of LLM is computed only using the response after the "</think>" token.

The results are presented in Table 5. We observe that our RPC method consistently outperforms PPL and SC across all datasets when using R1 LLMs. Moreover, the performance improvements persist even when the base LLM becomes stronger, demonstrating the potential of RPC to improve even stronger LLMs and its generalizability across diverse LLMs.

Table 5: Performance comparison on DeepSeek-R1-Distill-Qwen-7B model. The best performance is highlighted in **bold**. The results show that our RPC is compatible with thinking LLMs and consistently outperforms PPL and SC across all datasets.

| Method | MathOdyssey | AIME | MATH | OlympiadBench |
|--------|-------------|-------|-------|---------------|
| PPL | 60.04 | 72.92 | 81.81 | 21.65 |
| SC | 57.22 | 70.40 | 82.03 | 21.93 |
| RPC | **61.11** | **76.47** | **82.78** | **22.81** |

## D.4 Performance Comparison with Advanced Methods

In our main paper, we compare RPC with SC and PPL to demonstrate its effectiveness in bridging internal probability and self-consistency, which confirms our theoretical analysis. Here, we further combine RPC with two advanced methods: ESC [36] (an advanced version of SC) and Best-of-N (BoN) using the Qwen2.5-Math-PRM-7B reward model (RM) [69] (an advanced version of PPL that replaces internal probability with rewards from an external reward model) to verify the compatibility of RPC with state-of-the-art methods. Similar to Appendix D.3, we conduct experiments on the DeepSeek-R1-Distill-Qwen-7B model using 16 samples.

As shown in Table 6, RPC using ESC consistently outperforms both SC and ESC across all datasets, demonstrating its effectiveness when adapted to advanced methods. Similarly, the results in Table 7 show that RPC using RM consistently performs better than or comparable to SC and BoN using RM across all datasets. Overall, these experimental results demonstrate that RPC is compatible with state-of-the-art methods and consistently outperforms them across all datasets.

Table 6: Performance comparison with ESC method. The best performance is highlighted in **bold**.

| Method | MathOdyssey | AIME | MATH | OlympiadBench |
|---|---|---|---|---|
| Sc | 57.22 | 70.40 | 82.03 | 21.93 |
| ESC | 57.17 | 70.18 | 81.97 | 22.03 |
| Rpc using ESC | **61.03** | **76.26** | **82.74** | **22.81** |

Table 7: Performance comparison with BoN method using reward model. The best performance is highlighted in **bold**.

| Method | MathOdyssey | AIME | MATH | OlympiadBench |
|---|---|---|---|---|
| Sc | 57.22 | 70.40 | 82.03 | 21.93 |
| BoN using RM | **58.35** | 69.35 | 81.86 | 21.57 |
| Rpc using RM | **58.35** | **71.04** | **82.16** | **22.22** |

## D.5 Performance on Additional Reasoning Tasks

In our main paper, we demonstrate the effectiveness of Rpc on both mathematical reasoning and code generation tasks. Here, we further conduct experiments on two additional reasoning tasks: GPQA [47] and LogiQA [39]. GPQA is a multiple-choice benchmark covering biology, physics, and chemistry, while LogiQA is a benchmark testing logical reasoning abilities. Similar to Appendix D.3, we conduct experiments on the DeepSeek-R1-Distill-Qwen-7B model using 16 samples.

As shown in Table 8, we observe that our Rpc method consistently outperforms Ppl and Sc on both datasets. These experimental results demonstrate that the Rpc method is generalizable to additional reasoning tasks, showing its strong potential for application to diverse real-world reasoning scenarios.

Table 8: Performance comparison on GPQA and LogiQA benchmarks. The best performance is highlighted in **bold**.

| Method | GPQA | LogiQA |
|---|---|---|
| Ppl | 41.46 | 54.36 |
| Sc | 43.00 | 56.71 |
| Rpc | **44.09** | **58.42** |

## D.6 Explanation on Hyper-parameters

We would like to emphasize that the Rpc method introduces a Reasoning Pruning module that can automatically determine the pruning of reasoning paths without manually setting thresholds. Therefore, it avoids sensitivity issues associated with manual threshold setting and makes our Rpc a hyperparameter-free method, which is robust across diverse tasks, including mathematical reasoning, code generation, logical reasoning, etc.

Although our Rpc method does not explicitly require hyperparameter tuning, the optimization process of the Reasoning Pruning module still involves some parameter configurations, such as initialization and the range of distribution parameters. Therefore, to evaluate the robustness of Rpc, we conducted experiments analyzing the impact of different initialization methods and parameter bounds. Specifically, we tested the method with 10 random seeds under various configurations, including different initialization approaches (Fixed Init vs. Zero Init) and different bounds for the distribution parameters $w_1$ and $w_2$. The Fixed Init method and parameter bounds of $[0.2, 0.8]$ represent our default optimization settings for Rpc. Results presented in Table 9 demonstrate that Rpc maintains robust performance across different configurations. Note that these parameter configurations in our method do not require tuning in practical applications.

Table 9: Performance of RPC with different initialization methods and parameter bounds when automatically modeling internal LLM probability distributions.

|  | **Fixed Init** | **Zero Init** |
|---|---|---|
| $w \in [0.2, 0.8]$ | $31.620 \pm 0.754$ | $31.183 \pm 0.821$ |
| $w \in [0.15, 0.85]$ | $31.748 \pm 0.622$ | $32.108 \pm 0.685$ |
| $w \in [0.1, 0.9]$ | $31.722 \pm 0.577$ | $32.416 \pm 0.532$ |

### D.7 Explanation on Computational Overhead

In this section, we analyze the computational overhead of RPC compared to SC from both theoretical and practical perspectives.

From a theoretical standpoint, SC requires $O(n^2)$ time complexity for answer equivalence computation, where $n$ is the number of samplings. Our RPC method consists of two components: Reasoning Pruning with $O(k(m^2+n))$ time complexity and Perplexity Consistency with $O(n_p^2 + n_p \log n_p + n_p)$ time complexity, where $m = 5$ represents the number of mixture distribution parameters, $k$ denotes the optimization iteration count, and $n_p$ is the number of reasoning paths after pruning ($n_p \leq n$). The overall complexity simplifies to $O(25k + nk + n_p^2) \leq O(25k + nk + n^2)$, introducing only minimal computational overhead compared to SC.

From a practical perspective, experiments on MathOdyssey with 128 samplings show that SC takes 0.006s per question while RPC takes 0.036s. Both processing times are negligible compared to the LLM inference time required for sampling multiple reasoning paths. Moreover, RPC's slight overhead is well-justified by its ability to reduce the required number of samplings while maintaining performance, as demonstrated in Table 1, which addresses the primary computational bottleneck in reasoning tasks.

Overall, RPC provides an excellent computational trade-off, where the computational overhead of RP is exchanged for significant time savings achieved by reducing the number of LLM inferences.

## E Limitations and Future Work

In this paper, we introduce a theoretical framework for analysis the sampling-based test-time scaling methods of LLM reasoning and a new method based on our analyses. While our theoretical framework and approach offer several advantages, there remain some limitations as listed as follows:

1. Our theoretical framework is general to analysis sampling-based test-time scaling methods, however, in this paper, we only focus on the analysis on two typical methods as well as RPC.

2. Our proposed method is a post-hoc approach that relies solely on the sampled reasoning paths and selects answers from these samples. This design makes the method straightforward to use, as it does not require any modifications to the LLM architecture or its training process. However, the performance improvements are moderate compared to approaches that involve model training. Although integrating our method with trained LLMs may provide additional performance gains, it remains an open question how to utilize our insights to improve the training process of LLMs.

3. Current design of our method is relatively simple, which leaves considerable room for improvement. For example, the perplexity consistency component could be extended to include alternative probability measures, such as rewards from a reward model. Furthermore, the reasoning pruning module could be improved by incorporating more advanced techniques to increase its effectiveness.

We also provide the following directions for future work:

1. **Analyzing Test-Time Scaling Methods**: Our paper demonstrates that the theoretical framework is versatile enough to analyze two typical sampling-based test-time scaling methods. It can also be readily extended to evaluate other advanced methods derived from these two types, with strong potential to adapt to various test-time scaling strategies.

2. **Exploring Advanced Sampling Strategies**: Our theoretical results indicate that better convergence requires an effective sampling strategy to sample sufficiently diverse reasoning paths, which is not deeply investigated in this paper. Improved sampling strategies could be designed by drawing inspiration from our theoretical framework.

3. **Applying to Diverse Reasoning Tasks**: In our paper, we show that the Rpc method is effective on math, code, and logical reasoning tasks without exploiting task-specific properties. Task-specific methods could be developed based on our theoretical framework to achieve better performance (e.g., by considering the properties of $g(\cdot)$ in different reasoning tasks).

