# OpenReview forum: "A Theoretical Study on Bridging Internal Probability and Self-Consistency for LLM Reasoning"
_NeurIPS.cc/2025/Conference — NeurIPS 2025 poster_

### Official Review · Reviewer_CyKh · 2025-06-13

**Clarity:** 2
**Significance:** 2
**Originality:** 3
**Rating:** 4
**Confidence:** 3

**Summary:**

The paper titled "A Theoretical Study on Bridging Internal Probability and Self-Consistency for LLM Reasoning" presents a novel theoretical framework for analyzing test-time sampling strategies in large language models (LLMs) used for reasoning tasks. The authors introduce a method called Reasoning-Pruning Perplexity Consistency (RPC), which combines the strengths of self-consistency and perplexity-based methods to improve reasoning performance. The paper provides a detailed analysis of existing methods, identifies their limitations, and demonstrates through both theoretical and empirical results that RPC can significantly reduce reasoning error while maintaining high confidence reliability.

**Questions:**

1. Could the authors provide more details on the implementation of the mixture distribution modeling in the Reasoning Pruning component? How sensitive is the performance to the choice of distribution parameters?
2. Are there any other state-of-the-art test-time sampling methods that the authors plan to compare RPC with in future work? How does RPC compare to methods that involve fine-tuning or reinforcement learning for reasoning tasks?

**Ethical Concerns:**

["NO or VERY MINOR ethics concerns only"]

**Limitations:**

The computational complexity of RPC might limit its scalability to very large datasets or real-time applications. The authors should further investigate the trade-offs between performance and computational efficiency.

**Quality:**

3

**Strengths And Weaknesses:**

Strengths:
1. The paper introduces a comprehensive theoretical framework that decomposes reasoning error into estimation error and model error. This provides a principled way to analyze and understand the limitations of existing test-time sampling methods.
2. The proposed RPC method effectively combines the strengths of self-consistency and perplexity, achieving faster convergence rates and lower model error. This hybrid approach is supported by both theoretical analysis and empirical results.
3. The authors conduct extensive experiments on seven benchmark datasets, demonstrating that RPC outperforms existing methods in terms of reasoning accuracy, efficiency, and confidence reliability. The results are statistically significant and reproducible.
4. The paper provides detailed proofs and analysis of the theoretical results, which enhances the credibility and depth of the research.

Weaknesses:
1. Although the authors argue that the computational overhead of RPC is acceptable, the additional processing steps (e.g., mixture distribution modeling) could be a concern for real-time applications where inference speed is critical.
2. The effectiveness of RPC relies on the quality of the underlying LLM. If the LLM itself has significant reasoning errors or biases, the improvements brought by RPC might be limited.

---

> ### Author Rebuttal · Authors · 2025-07-31
>
> **Dear Reviewer CyKh:**
>
> Thank you for your valuable feedback on our paper. We address your concerns as follows.
>
> ### **Weakness 1 & Limitations: Computational Overhead of RPC**
>
> We appreciate your concern about computational efficiency. We would like to emphasize that the primary computational bottleneck in reasoning tasks lies in the time consumed by multiple LLM inferences.
> As demonstrated in Table 1, RPC can achieve the same or better performance as SC with fewer samplings, showing strong potential to reduce the computational overhead of LLM inference.
> Therefore, RPC actually provides an excellent computational trade-off, where the minimal computational overhead of RPC is exchanged for significant time savings by reducing the number of required LLM inferences.
>
> As detailed in Appendix D.5, our thorough efficiency analysis shows that RPC maintains a time complexity comparable to SC, with only a negligible overhead of 0.03s per problem when processing 128 samplings on the MathOdyssey dataset. Therefore, RPC demonstrates practicality for real-time applications where inference speed is critical, as it achieves better performance with fewer LLM samplings while introducing only negligible computational overhead compared to existing methods.
>
> ### **Weakness 2: Performance of RPC regarding to underlying LLM**
>
> We appreciate your insightful concern about the dependency on the underlying LLM quality. However, we would like to highlight that this dependency affects all comparison methods (SC and PPL) equally, and we compare the performance using the same LLM to ensure the fairness of experiments. Furthermore, our experiments in Table 3 and Figure 4 demonstrate that RPC consistently delivers performance improvements across different model architectures and sizes, indicating the robustness of our approach.
>
> From a theoretical perspective, our framework explicitly addresses this concern by decomposing reasoning error into two components: model error (inherent to the specific LLM) and estimation error (which can be optimized by the algorithms). This decomposition allows us to isolate the impact of the underlying model quality (model error) and focus on improving the estimation process (estimation error). RPC is specifically designed to minimize this estimation error component, making it effective regardless of the base model, as long as the model provides meaningful reasoning signals.
>
> ### **Question 1: Details of Reasoning Pruning**
>
> Thank you for the question about our implementation details of Reasoning Pruning.
> Reasoning Pruning is a parameter-free method that automatically determines the pruning of reasoning paths without requiring manual setting of a threshold.
> Its core idea is modeling LLM probabilities across multiple samplings as a mixture of two distributions: a normal confidence distribution and an extremely low confidence distribution. This approach allows us to automatically identify and prune low-quality reasoning paths without requiring manual threshold setting.
> For implementation details, we model the LLM probability distribution using the mixture model specified in the equation below Line 211. We estimate the distribution parameters ($w_1$, $w_2$, $k_1$, $k_2$, $\lambda_1$, $\lambda_2$) using maximum likelihood estimation implemented with SciPy's optimization tool. This process is described in Algorithm 1 (Lines 5-7) in Appendix B.
>
> A key advantage of our approach is that all distribution parameters are automatically estimated from the probability of LLM samplings rather than being manually configured, eliminating sensitivity concerns. Our extensive experiments across various models and diverse datasets consistently demonstrate the robustness of this automatic approach.
>
> ### **Question 2: Future Work of RPC**
>
> Thank you for your suggestions. We plan to compare RPC with more methods using more LLMs in future work to evaluate its effectiveness. The following preliminary experiments verify the feasibility of this plan.
>
> We evaluate RPC using LLM with the long CoT reasoning capability [2], i.e., DeepSeek-R1-Distill-Qwen-7B [1], to determine whether RPC can enhance performance when combined with the SOTA test-time sampling technique. We compare the best-of-N method (BoN), which selects the samplings according to the average process reward scores generated by Qwen2.5-Math-PRM 7B [4]. Additionally, we compare an advanced self-consistency method, i.e., ESC [3]. Due to computational resource constraints, we only evaluate and report two relatively small datasets, i.e., MathOdessey and AIME 1983-2024, with a maximum of 16 samplings per problem as follows. The results demonstrate that our RPC consistently outperforms all comparison methods, demonstrating its effectiveness.
>
> |  | MathOdessey | AIME 1983-2024 |
> | --- | --- | --- |
> | PPL | 60.04 | 72.92 |
> | SC | 57.22 | 70.40 |
> | BoN using RM [4] | 58.35 | 69.35 |
> | ESC [3] | 57.17 | 70.18 |
> | RPC | **61.11** | **76.47** |
>
> Regarding fine-tuning or reinforcement learning models, we would like to clarify that RPC is a post-hoc method that can be easily integrated with a trained model or sampled reasoning paths. For example, we have demonstrated that RPC can be directly applied to the DeepSeek-R1-Distill-Qwen-7B model to further improve the performance in the previous table.
>
> Moreover, RPC can further enhance advanced methods. For instance, ESC can be upgraded through RPC (RPC using ESC), and RPC can also leverage reward scores generated by a reward model (RPC using RM). The results, presented in the following two tables, show that both RPC using ESC and RPC using RM methods outperform the SC method while maintaining comparable or superior performance to their original counterparts, demonstrating the effectiveness of RPC.
>
> |  | MathOdessey | AIME 1983-2024 |
> | --- | --- | --- |
> | SC | 57.22 | 70.40 |
> | ESC [3] | 57.17 | 70.18 |
> | RPC using ESC [3] | **61.03** | **76.26** |
>
> |  | MathOdessey | AIME 1983-2024 |
> | --- | --- | --- |
> | SC | 57.22 | 70.40 |
> | BoN using RM [4] | **58.35** | 69.35 |
> | RPC using RM [4] | **58.35** | **71.04** |
>
> ### **Reference**
>
> [1] DeepSeek-R1: Incentivizing Reasoning Capability in LLMs via Reinforcement Learning. Arxiv 2025.
>
> [2] Towards Reasoning Era: A Survey of Long Chain-of-Thought for Reasoning Large Language Models. Arxiv 2025.
>
> [3] Escape Sky-high Cost: Early-stopping Self-Consistency for Multi-step Reasoning. ICLR 2024.
>
> [4] The Lessons of Developing Process Reward Models in Mathematical Reasoning. Arxiv 2025.

---

> ### Author Response · Authors · 2025-08-06
>
> Dear Reviewer CyKh,
>
> Thank you for your insightful feedback, as well as the time and effort you have dedicated to reviewing our paper. We have carefully considered your questions and suggestions and have addressed them in our response. We sincerely appreciate your constructive comments, which have helped strengthen our paper.
>
> As the discussion phase is nearing its end, we would be most grateful if you could inform us whether you have any additional questions or further suggestions.
>
> Best regards,
>
> The authors

---

> > ### Comment · Area_Chair_5wTM · 2025-08-09
> >
> > Dear Reviewer CyKh,
> >
> > Can you please take a look at the authors' rebuttal and respond accordingly?
> >
> > Thanks, AC

---

### Official Review · Reviewer_CQT8 · 2025-07-04

**Clarity:** 3
**Significance:** 2
**Originality:** 2
**Rating:** 4
**Confidence:** 3

**Summary:**

Test-time sampling is a hot topic to improve the reasoning performance of large language models (LLMs). In this paper, it gives the theoretical analysis for test-time sampling and two dominant paradigms self-consistency and perplexity are analyzed. Meanwhile, the limitations are also revealed. Then RPC is introduced to leverage theoretical insights through two key components: Perplexity Consistency and Reasoning Pruning. The experiments show that the sampling costs are signifcicantly reduced and the confidence reliability is enhanced.

**Questions:**

1.The challenges to analyze the self-consistency and perplexity for the test-time sampling should be further explained since it is the motivation of the paper and can improve the quality of the paper well.
2.Since test-time sampling is a hot topic, the works that are published in recent two years should be compared and analyzed.
3. In this paper, a framework is designed. However, I wonder whether the frameworks can be integated into the previous works and the framework should be further verifed.

**Ethical Concerns:**

["NO or VERY MINOR ethics concerns only"]

**Final Justification:**

Most of my concerns have been addressed and the experiments seem to be promising.

**Limitations:**

The authors state that there are few works to give the theoretical analysis. However, the challenges to give the theoretical analysis should be deeply expained since it is the foundation of the paper.

**Paper Formatting Concerns:**

I have no formatting concerns.

**Quality:**

3

**Strengths And Weaknesses:**

Strengths:
1.The paper is well organized and it is easy to follow.
2.The theoretical analysis can expain the reasoning of LLM well. The experiments are well designed.

Weakeness:
1.The challenges to analyze the self-consistency and perplexity for the test-time sampling should be further explained.
2.More SOTA methods with test-time sampling should be comapred.
3.The framework should be further verifed.

---

> ### Author Rebuttal · Authors · 2025-07-31
>
> **Dear Reviewer CQT8:**
>
> Thank you for your valuable feedback on our paper. We address your concerns as follows.
>
> ### **Weakness 1 & Question 1 & Limitations: Challenges of Theoretical Analysis**
>
> Thank you for your insightful question regarding the challenges of theoretical analysis and our motivation. We would like to clarify the challenges of theoretical analysis as follows.
>
> First, a significant gap exists in the theoretical understanding of current methods like Self-Consistency (SC) and Perplexity (PPL). There is no conclusive explanation for their respective strengths or the specific conditions under which each excels. This gap motivated us to develop a robust theoretical framework. Our goal is to not only explain their empirical performance but also to identify their inherent disadvantages and, crucially, devise a method (RPC) that synergistically combines their strengths.
>
> Second, the fundamental difficulty in combining SC and PPL lies in their distinct methodological underpinnings. Self-Consistency (SC) operates as a sampling-based estimation method, approximating the true probability distribution by aggregating multiple reasoning paths. In contrast, Perplexity (PPL) is a direct estimation method that calculates probabilities straight from the model. The disparate principles governing these two approaches make the theoretical analysis of their combination particularly challenging.
>
> ### **Weakness 2 & Question 2: Comparison with Recent Methods**
>
> Thank you for your suggestion. We evaluate each method using a recent LLM with the long CoT reasoning capability [2], i.e., DeepSeek-R1-Distill-Qwen-7B [1], to determine whether RPC can enhance performance when combined with SOTA test-time sampling technique. We also compare the best-of-N method (BoN) using a reward model, which selects the samplings according to average process reward scores generated by Qwen2.5-Math-PRM 7B [4]. Additionally, we compare an advanced self-consistency method, i.e., ESC [3]. Due to computational resource constraints, we only evaluate two relative small datasets, i.e., MathOdessey and AIME 1983-2024, with a maximum of 16 samplings per problem as follows.
>
> |  | MathOdessey | AIME 1983-2024 |
> | --- | --- | --- |
> | PPL | 60.04 | 72.92 |
> | SC | 57.22 | 70.40 |
> | BoN using RM [4] | 58.35 | 69.35 |
> | ESC [3] | 57.17 | 70.18 |
> | RPC | **61.11** | **76.47** |
>
> The results demonstrate that our RPC consistently outperforms all comparison methods, demonstrating its effectiveness.
>
> ### **Weakness 3 & Question 3: Further Verification on Theoretical Framework**
>
> Thank you for your suggestion. Our framework is generally compatible with extended versions of both PPL and SC methods, meaning the PPL or SC functions can be readily substituted with more advanced alternatives.  For instance, consider integrating ESC [3], an advanced self-consistency method. Our theoretical framework readily extends to this by analyzing the error between the ground-truth probability and the estimated probability derived from ESC's specific computation functions. In this scenario, our theoretical analysis (Proposition 2 and Theorem 4) remains valid, and the new SC method does not alter our proposed framework.
>
> To empirically validate this,  our experimental results below demonstrate that RPC remains effective when the SC function is replaced by ESC or when the PPL function is replaced by reward functions from an external Reward Model (RM). Both RPC using ESC and RPC using RM methods outperform the baseline SC method while maintaining comparable or superior performance relative to their respective counterparts. These findings emphasize the effectiveness of RPC and empirically verify our theoretical framework.
>
> |  | MathOdessey | AIME 1983-2024 |
> | --- | --- | --- |
> | SC | 57.22 | 70.40 |
> | ESC [3] | 57.17 | 70.18 |
> | RPC using ESC [3] | **61.03** | **76.26** |
>
> |  | MathOdessey | AIME 1983-2024 |
> | --- | --- | --- |
> | SC | 57.22 | 70.40 |
> | BoN using RM [4] | **58.35** | 69.35 |
> | RPC using RM [4] | **58.35** | **71.04** |
>
> We believe our theoretical framework provides a foundation that can help future researchers identify the limitations of various test-time sampling methods and inspire meaningful improvements to these approaches.
>
> ### **Reference**
>
> [1] DeepSeek-R1: Incentivizing Reasoning Capability in LLMs via Reinforcement Learning. Arxiv 2025.
>
> [2] Towards Reasoning Era: A Survey of Long Chain-of-Thought for Reasoning Large Language Models. Arxiv 2025.
>
> [3] Escape Sky-high Cost: Early-stopping Self-Consistency for Multi-step Reasoning. ICLR 2024.
>
> [4] The Lessons of Developing Process Reward Models in Mathematical Reasoning. Arxiv 2025.

---

> ### Author Response · Authors · 2025-08-06
>
> Dear Reviewer CQT8,
>
> Thank you for your insightful feedback, as well as the time and effort you have dedicated to reviewing our paper. We have carefully considered your questions and suggestions and have addressed them in our response. We sincerely appreciate your constructive comments, which have helped strengthen our paper.
>
> As the discussion phase is nearing its end, we would be most grateful if you could inform us whether you have any additional questions or further suggestions.
>
> Best regards,
>
> The authors

---

> > ### Comment · Area_Chair_5wTM · 2025-08-09
> >
> > Dear Reviewer CQT8,
> >
> > Can you please take a look at the authors' rebuttal and respond accordingly?
> >
> > Thanks,
> > AC

---

> > > ### Comment · Reviewer_CQT8 · 2025-08-09
> > > **response**
> > >
> > > I have read the rebuttal and I will keep my score.

---

> > > > ### Author Response · Authors · 2025-08-09
> > > >
> > > > Dear Reviewer CQT8,
> > > >
> > > > We sincerely appreciate the time and effort you have dedicated to reviewing our paper, as well as your thoughtful comments.
> > > >
> > > > We are grateful that you can respond to our rebuttal on the last day of the author-reviewer discussions. Your insightful suggestions have been invaluable in helping us improve the quality of our paper.
> > > >
> > > > Best regards,
> > > >
> > > > The authors

---

### Official Review · Reviewer_bSKW · 2025-07-05

**Clarity:** 3
**Significance:** 3
**Originality:** 3
**Rating:** 4
**Confidence:** 3

**Summary:**

This paper lays out a new way to think about test-time sampling in large language model reasoning, splitting the error into two parts: estimation error from sampling and model error from the LLM itself. It introduces a method called Reasoning-pruning Perplexity Consistency (RPC), which blends the strengths of self-consistency and perplexity while trimming less likely reasoning paths to cut sampling needs by half. Through experiments on seven datasets, including math and coding tasks, RPC shows it can match or beat existing methods in accuracy while making confidence estimates more reliable.

**Questions:**

1. Have you considered testing RPC on tasks beyond math and coding, like commonsense reasoning or text-based decision-making? Showing it works across diverse domains would make the method’s impact more convincing and could sway my view on its significance.
2. What’s the computational trade-off for RPC, especially with the Weibull distribution fitting and pruning steps? Providing runtime or resource comparisons with baselines would clarify its feasibility and could strengthen my confidence in its practicality.
3. How sensitive is RPC’s performance to the pruning threshold τ? An ablation study showing how different τ values affect results would help me trust the method’s robustness and potentially improve my quality assessment.

**Ethical Concerns:**

["NO or VERY MINOR ethics concerns only"]

**Final Justification:**

Thank you to the authors for their thoughtful and insightful responses. I maintain my positive score, and I hope this paper will be accepted.

**Limitations:**

yes

**Quality:**

3

**Strengths And Weaknesses:**

Strengths:
- The theoretical framework is a standout, offering a clear lens to analyze sampling strategies with solid mathematical backing, including detailed proofs tucked in the appendix.
- The writing flows logically, guiding readers from problem setup to RPC’s design and results, with figures and tables that make the method’s benefits easy to grasp.
- By slashing sampling requirements by 50% without sacrificing performance, RPC tackles a real bottleneck in deploying LLMs efficiently, which could have broad practical impact.
- RPC’s blend of perplexity, self-consistency, and a pruning mechanism feels fresh, cleverly building on prior work to address specific shortcomings.

Weaknesses:
- The experiments stick to math and coding datasets, so it’s unclear how well RPC would handle other reasoning tasks like commonsense or creative problems, limiting its proven scope.
- The explanation of the Weibull distribution for pruning is a bit thin, which might leave readers puzzled about how it’s implemented or why it’s chosen.

---

> ### Author Rebuttal · Authors · 2025-07-31
>
> **Dear Reviewer bSKW:**
>
> Thank you for your valuable feedback on our paper. We address your concerns as follows.
>
> ### **Weakness 1 & Question 1: Additional Evaluation Benchmarks**
>
> Thank you for your suggestion. To address your concern about the generalizability of our method beyond mathematical and coding tasks, we conducted additional experiments on diverse reasoning domains. Specifically, we evaluated RPC on GPQA [2] (a multiple-choice benchmark covering biology, physics, and chemistry) and LogiQA [3] (a benchmark testing logical reasoning abilities) using the DeepSeek-R1-Distill-Qwen-7B model [1].
>
> |  | GPQA | LogiQA |
> | --- | --- | --- |
> | PPL | 41.46 | 54.36 |
> | SC | 43.00 | 56.71 |
> | RPC | **44.09** | **58.42** |
>
> These additional experimental results across diverse reasoning benchmarks provide strong empirical support for our theoretical framework and demonstrate the broad applicability of RPC beyond mathematical and coding tasks, addressing your concern about the generalizability of our method.
>
> ### **Weakness 2: Explanation of Reasoning Pruning**
>
> Thank you for raising this question. Our core idea is to model the probability distribution of LLM samplings as a mixture of two distributions: one for high-confidence reasoning paths and another for low-confidence paths. This allows us to automatically identify which samplings belong to the low-probability distribution for pruning, thereby eliminating the need for manually determined thresholds and their associated sensitivity issues.
>
> Regarding implementation details, we selected the Weibull distribution as our base distribution specifically because it effectively captures low confidence values, which is essential for our pruning. The distribution parameters ($w_1$, $w_2$, $k_1$, $k_2$, $\lambda_1$, $\lambda_2$) are estimated using maximum likelihood estimation via SciPy's optimization tools, as detailed in Algorithm 1 (Lines 5-7) in Appendix B.
> As mentioned in our Limitations in Appendix E, our current implementation of Reasoning Pruning is intentionally straightforward to validate our theoretical framework, with potential for further enhancement through alternative distributions or more advanced techniques.
>
> We will revise our paper to provide clearer explanations of both the implementation details and motivation.
>
> ### **Question 2: Computational Overhead of RPC**
>
> We appreciate your concern about the computational trade-offs. We would like to clarify that the primary computational bottleneck in reasoning tasks lies in the time consumed by multiple LLM inferences. As demonstrated in Table 1, RPC can achieve the same performance as SC with fewer samplings, showing strong potential to reduce the LLM inference overhead. Therefore, RPC actually provides an excellent computational trade-off, where the minimal computational overhead of RPC is exchanged for significant time savings by reducing the number of required LLM inferences.
>
> We provide a detailed analysis of computational overhead in Appendix D.5.
> Moreover, taking the LogiQA benchmark mentioned above as an example, the average time for one LLM inference is approximately 0.7s per question.
> The computational overhead of SC is 0.0001s per question, while the computational overhead of RPC is 0.008s per question, when dealing with 16 samplings. Therefore, the computational overhead of either method is negligible compared to the LLM inference time. Even saving a single LLM inference completely offsets the additional computational overhead of any method.
>
> We will revise our paper to clarify the computational issue, and we hope our clarification will address your concern.
>
> ### **Question 3: Questions about Pruning Threshold $\tau$**
>
> We thank you for highlighting this important point.
> To clarify, our Reasoning Pruning is designed as a parameter-free method that automatically determines the pruning of reasoning paths without requiring manual setting of a threshold $\tau$. This determination is based on the probability distribution of reasoning paths, as explained in our response to Weakness 2.
> This automatic determination eliminates any sensitivity issues that would typically arise from manual threshold setting.
> The consistently superior performance of RPC across diverse tasks and different models demonstrates the effectiveness and robustness of our automatic approach in the Reasoning Pruning component.
>
> In the revised paper, we will clarify and emphasize the advantage of the Reasoning Pruning module for automatic pruning.
>
> ### **Reference**
>
> [1] DeepSeek-R1: Incentivizing Reasoning Capability in LLMs via Reinforcement Learning. Arxiv 2025.
>
> [2] GPQA: A Graduate-Level Google-Proof Q&A Benchmark. COLM 2024.
>
> [3] LogiQA: A Challenge Dataset for Machine Reading Comprehension with Logical Reasoning. IJCAI 2020.

---

> > ### Comment · Reviewer_bSKW · 2025-08-05
> > **Reviewer Response**
> >
> > Thank you to the authors for the detailed rebuttal, which has addressed all of my concerns. This is a good paper, and I will maintain my positive score.

---

> ### Author Response · Authors · 2025-08-06
>
> Dear Reviewer bSKW,
>
> We sincerely appreciate the time and effort you have dedicated to reviewing our paper, as well as your thoughtful comments. We are pleased to hear that our responses have adequately addressed all of your concerns. Your insightful suggestions have been invaluable in helping us improve the quality of our paper.
>
> Moreover, we look forward to any further discussion to enhance our paper. If you have any questions or suggestions to help us improve our paper, please feel free to ask, and we will be delighted to address them and incorporate your valuable feedback.
>
> Best regards,
>
> The authors

---

### Official Review · Reviewer_m4pU · 2025-07-07

**Clarity:** 3
**Significance:** 3
**Originality:** 3
**Rating:** 4
**Confidence:** 2

**Summary:**

They present a theoretical analysis of several methods for aggregating multiple LLM samples: perplexity based and consistency based. They break the model's error into both an estimation error component and a model error component. Using their framework, they then propose two improvements which improve both the estimation/model error theoretically: perplexity consistency and reasoning pruning. They show that combining these techniques can also empirically yield improvements on best of N on several math benchmarks with open LLMs.

**Questions:**

See weaknesses.

**Ethical Concerns:**

["NO or VERY MINOR ethics concerns only"]

**Final Justification:**

I believe the additional experiments they provide help give further empirical evidence for their method, so I am willing to raise my score to a 4.

**Limitations:**

I think they could evaluate on more benchmarks and using more LLMs to strengthen their empirical claims. The gains observed are not that huge.

**Paper Formatting Concerns:**

None!

**Quality:**

3

**Strengths And Weaknesses:**

Strengths:
* They present an interesting theoretically framework for understanding the efficacy of different best-of-N test time compute aggregation methods.
* Their theoretical framework allows them to come up with an improvement over existing techniques which yields empirical gains.
* The paper is largely well written
* They carry out reasonably thorough empirical experiments
* I like the use of both accuracy and calibration as metrics for evaluation

Weaknesses:
* The gains from their new method are quite small on most of the tasks they experiment with, with the exception of MathOdessey, which shows much more substantial gains. Why is that? Is there something special about that particular text which lends itself to this method?
* The ideas derived from their theory feel sort of obvious. I guess it's nice to have all the theory, but if you asked me to list off 20 ideas for ways one could potentially improve over majority voting, I would probably come up with these, and then I would be able to pick them with empirical validation from there. This isn't to necessarily take away from the work, but at points the theory feels a little excessive for what it is helping to deliver in the paper.
* They do all their experiments using two LLaMA models, but could demonstrate the generality of their approach better by showing improvements using more open LLMs.

---

> ### Author Rebuttal · Authors · 2025-07-31
>
> **Dear Reviewer m4pU:**
>
> Thank you for your valuable feedback on our paper. We address your concerns as follows.
>
> ### **Weakness 1: Explanation for the MathOdyssey Dataset**
>
> Technically, RPC effectively integrates the strengths of both SC and PPL methods. When these two approaches produce good and complementary results, RPC can leverage this synergy to achieve substantial performance improvements. In the case of the MathOdyssey dataset, Figure 2(b) illustrates this complementarity: as the number of samplings increases, PPL demonstrates a large initial advantage, while SC performance improves later. This complementarity enables RPC to exploit the strengths of both methods, resulting in the significant performance gains observed.
>
> ### **Weakness 1 & Limitations: Magnitude of Performance Gains**
>
> We would like to clarify that RPC's performance gains should be considered in the proper context. RPC is a general framework with two key features: it is a training-free method that needs no extra data collection or model fine-tuning; and it is a post-hoc method, meaning any PPL or SC functions can be directly integrated. Given these points, even modest but consistent accuracy improvements across diverse, challenging reasoning benchmarks are quite significant. Additionally, our approach offers several advantages beyond accuracy:
> (1) **Efficiency**: As demonstrated by our theoretical results and confirmed in Table 1, RPC achieves faster estimation error convergence, potentially reducing the computational costs;
> (2) **Calibration**: As demonstrated in Table 2, RPC also produces more reliable confidence scores, enhancing the reliability of LLM prediction.
>
> In summary, when evaluated comprehensively across accuracy, efficiency, and calibration, our empirical and theoretical results strongly support the advantages of the RPC approach.
>
> ### **Weakness 2: Importance of Theoretical Results**
>
> We would like to emphasize that the empirical exploration-guided method and the theoretical analysis-guided method represent two distinct research methodologies with different strengths.
> While we acknowledge that effective algorithms can emerge through empirical exploration, theoretical analysis provides unique advantages in systematically identifying the fundamental limitations of existing algorithms and developing principled solutions.
>
> As presented in our paper, our theoretical results reveal the fundamental limitations of SC and PPL methods. We then derive potential solutions by combining these methods and provide theoretical results that reveal both advantages and challenges of this approach. RPC serves as a direct implementation of our theoretical insights, which empirically validates our theoretical framework and demonstrates the effectiveness of our proposed solution. Therefore, we believe that beyond the specific algorithm we proposed, our theoretical results provide valuable insights and make significant contributions to the field.
>
> ### **Weakness 3 & Limitations: Experiments with Additional LLMs**
>
> Thank you for your suggestion. To address this concern, we conducted additional experiments using DeepSeek-R1-Distill-Qwen-7B [1], a recent LLM with long CoT reasoning capability [2]. Due to computational resource constraints, we focused on two relatively small datasets: MathOdyssey and AIME 1983-2024, using 16 samplings per question for each method. We evaluated both accuracy and calibration metrics across these datasets.
> The results consistently show that, when a better base model is adopted, RPC still outperforms baseline methods while maintaining good calibration, which further validates the generalizability of our approach across different models.
>
> |  | MathOdyssey Acc. | MathOdyssey ECE | AIME Acc. | AIME ECE |
> | --- | --- | --- | --- | --- |
> | PPL | 60.04 | 20.2 | 72.92 | 4.3 |
> | SC | 57.22 | **11.4** | 70.40 | 5.3 |
> | RPC | **61.11** | **11.4** | **76.47** | **2.5** |
>
> ### **Limitations: Experiments with Additional Benchmarks**
>
> Thank you for your suggestion. We further validate our proposed method across diverse reasoning domains. Specifically, we evaluate RPC on GPQA [3] (a multiple-choice benchmark covering biology, physics, and chemistry) and LogiQA [4] (a benchmark testing logical reasoning abilities) using the DeepSeek-R1-Distill-Qwen-7B model [1].
>
> We report the performance of PPL, SC, and RPC using 16 samplings per question. Additionally, we evaluate RPC with only 10 samplings (denoted as RPC*), which reduces LLM inference costs by 37.5% compared to other methods. The results demonstrate that RPC consistently outperforms baseline methods across different reasoning tasks.
> Notably, even with fewer samplings, RPC* still achieves better performance than SC and PPL methods (which use 16 samplings), demonstrating both the effectiveness and efficiency of our approach and its broad applicability.
>
> |  | GPQA | LogiQA |
> | --- | --- | --- |
> | PPL | 41.46 | 54.36 |
> | SC | 43.00 | 56.71 |
> | RPC* (w/ 10 samplings) | 43.58 | 57.59 |
> | RPC | **44.09** | **58.42** |
>
> ### **Reference**
>
> [1] DeepSeek-R1: Incentivizing Reasoning Capability in LLMs via Reinforcement Learning. Arxiv 2025.
>
> [2] Towards Reasoning Era: A Survey of Long Chain-of-Thought for Reasoning Large Language Models. Arxiv 2025.
>
> [3] GPQA: A Graduate-Level Google-Proof Q&A Benchmark. COLM 2024.
>
> [4] LogiQA: A Challenge Dataset for Machine Reading Comprehension with Logical Reasoning. IJCAI 2020.

---

> ### Author Response · Authors · 2025-08-06
>
> Dear Reviewer m4pU,
>
> Thank you for your insightful feedback, as well as the time and effort you have dedicated to reviewing our paper. We have carefully considered your questions and suggestions and have addressed them in our response. We sincerely appreciate your constructive comments, which have helped strengthen our paper.
>
> As the discussion phase is nearing its end, we would be most grateful if you could inform us whether you have any additional questions or further suggestions.
>
> Best regards,
>
> The authors

---

> > ### Comment · Area_Chair_5wTM · 2025-08-09
> >
> > Dear Reviewer m4pU,
> >
> > Can you please take a look at the authors' rebuttal and respond accordingly?
> >
> > Thanks,
> > AC

---

### Author Response · Authors · 2025-08-08
**General Response**

We would like to thank the reviewers for their insightful feedback, as well as the time and effort they have dedicated to reviewing our paper. Below, we summarize our responses to the major issues raised.

Reviewers m4pU and bSKW requested more experimental results to show the generalization of our RPC method to additional benchmarks across different domains and more LLMs. To address this issue, **we have provided experimental results using the recent LLM with long CoT capacity [1] and reported performance on GPQA [2] and LogiQA [3] datasets covering biology, physics, chemistry, and logical reasoning domains** (detailed in our responses to reviewers m4pU and bSKW). The results demonstrate that RPC generalizes well to more benchmarks and LLMs.

Reviewers CQT8 and CyKh requested more comparisons with state-of-the-art methods. In our response, **we have provided additional comparison results with recent methods, such as ESC [4] and BoN using the recent reward model [5], showing the effectiveness of our RPC** (detailed in our responses to reviewers CQT8 and CyKh). We further demonstrate that RPC can improve the performance of recent methods, **highlighting RPC’s advantage of being integrable with other methods**.

Reviewers bSKW and CyKh requested further explanation about our Reasoning Pruning module and the computational costs of RPC. In our responses, we have provided a detailed implementation of Reasoning Pruning and highlighted **its advantage of automatically pruning reasoning paths without requiring manual threshold setting**. Regarding computational costs, we show that **RPC provides an excellent trade-off, where minimal computational overhead is exchanged for significant time savings by reducing the number of required LLM inferences**.

Reviewers m4pU and CQT8 requested further discussion about our theoretical framework. In our response to m4pU, **we highlight the significance of our theoretical results, which help systematically identify the fundamental limitations of existing algorithms and guide the development of principled solutions**. In our response to CQT8, **we further discuss the challenges of our analysis in combining two distinct methodological underpinnings and provide additional verification of our theoretical framework**.

We sincerely appreciate the reviewers' constructive comments, which have helped strengthen our paper. Since the discussion phase ends today, we would be grateful to know whether we have adequately addressed the reviewers' questions and concerns or if they have any further feedback.

### **Reference**

[1] DeepSeek-R1: Incentivizing Reasoning Capability in LLMs via Reinforcement Learning. Arxiv 2025.

[2] GPQA: A Graduate-Level Google-Proof Q&A Benchmark. COLM 2024.

[3] LogiQA: A Challenge Dataset for Machine Reading Comprehension with Logical Reasoning. IJCAI 2020.

[4] Escape Sky-high Cost: Early-stopping Self-Consistency for Multi-step Reasoning. ICLR 2024.

[5] The Lessons of Developing Process Reward Models in Mathematical Reasoning. Arxiv 2025.

---

### Note · Authors · 2025-08-12

**Dear SAC, AC, and Reviewers,**

We thank all reviewers and the AC for their constructive feedback. We are encouraged by the positive comments about the strengths of our paper in the review process:

- **The theoretical framework is interesting and stands out in understanding the LLM reasoning well**. (Reviewers m4pU, bSKW, CQT8, CyKh)
- **RPC delivers superior accuracy, efficiency, and reliability**, backed by statistically robust results. (Reviewers bSKW, CyKh)
- **The paper is well-written and logically structured**, with effective visualization of figures and tables. (Reviewers m4pU, bSKW, CQT8)
- **The experiments are well-designed and reasonably thorough**, using multiple metrics for evaluation. (Reviewers m4pU, CyKh)

Further, we summarize our response to reviews to clarify some misconceptions and strengthen our claims:

- **[Reasoning Pruning and Computational Overhead]**

    **Reasoning Pruning (RP) can automatically determine pruning of reasoning paths without manually setting the threshold**, avoiding sensitivity issues. **The additional computational overhead of RP is very small** and can be ignored. Moreover, **RPC provides an excellent computational trade-off**, where minimal computational overhead of RP is exchanged for significant time savings by reducing the number of required LLM inferences.

- **[Clarification of Theoretical Framework]**

    **The significance of our theoretical result is highlighted by revealing limitations of existing algorithms and guiding future developments**, while **its challenge lies in combining two distinct methodological underpinnings**. Moreover, we provide a sketch on how to integrate our analysis with other methods and conduct empirical verification to further verify our framework.

- **[Extended Experimental Results]**

    We provide experimental results on a recent LLM on math benchmarks and benchmarks covering biology, physics, chemistry, and logical reasoning domains, **demonstrating our proposal can generalize to more LLMs and more benchmarks**. We provide the comparison with methods such as the BoN method using a reward model and the ESC method, as well as the combination of RPC with these methods, to **prove RPC's effectiveness and flexibility**.

To summarize, we propose **a theoretical framework to understand test-time sampling for LLM reasoning** and **an effective, efficient, and theoretically consistent method across many benchmarks and models**.

Best regards,

The Authors

---

### Decision · Program_Chairs · 2025-09-17

**Decision:**

Accept (poster)

**Comment:**

This paper presents an interesting theoretical analysis on various methods of ensembling LLM output, which leads to design of Perplexity Consistency and Reasoning Pruning components. Through the discussion in Rebuttal phase, the reviewers reach consensus on accepting this paper.